# The Diversity and Taxonomy of Thelephoraceae (Basidiomycota) with Descriptions of Four Species from Southwestern China

**DOI:** 10.3390/jof10110775

**Published:** 2024-11-07

**Authors:** Xiaojie Zhang, Fulei Shi, Sicheng Zhang, Md. Iqbal Hosen, Changlin Zhao

**Affiliations:** 1Yunnan Provincial Key Laboratory for Conservation and Utilization of In-Forest Resource, The Key Laboratory of Forest Resources Conservation and Utilization in the Southwest Mountains of China Ministry of Education, Southwest Forestry University, Kunming 650224, China; fungixiaojie@swfu.edu.cn (X.Z.); fungifuleishi@163.com (F.S.); fungizhangsicheng@163.com (S.Z.); iqbalmyco@gmail.com (M.I.H.); 2College of Forestry, Southwest Forestry University, Kunming 650224, China

**Keywords:** biodiversity, classification, phylogenetic analyses, taxonomy, wood-inhabiting fungi, Yunnan province

## Abstract

Taxonomy plays a central role in understanding the diversity of life, translating the products of biological exploration and discovery specimens and observations into systems of names that settle a “classification home” to taxa. The ectomycorrhizal basidiomycetes family Thelephoraceae has been understudied in subtropical ecosystems. Many species of Thelephoraceae are important edible and medicinal fungi, with substantial economic value. Four new species, *Thelephora resupinata*, *T. subtropica*, *T. yunnanensis,* and *Tomentella tenuifarinacea*, are proposed based on a combination of the morphological features and molecular evidence. *Thelephora resupinata* is characterized by the resupinate basidiomata having a tuberculate pileal surface hymenial, and the presence of the subglobose to globose basidiospores (9–12 × 7–9 µm). *T. subtropica* is solitary coriaceous infundibuliform gray-brown basidiomata with a presence of the subclavate basidia and subglobose to globose basidiospores (6–8 × 5–7 µm). *T. yunnanensis* is typical of the laterally stipitate basidiomata having a smooth, umber to coffee hymenial surface, a monomitic hyphal system with clamped generative hyphae, and the presence of the subglobose basidiospores (7–10 × 6–8 µm). *Tomentella tenuifarinacea* is typical of the arachnoid basidiomata having a smooth, gray, or dark gray hymenial surface, a monomitic hyphal system with clamped generative hyphae, and the presence of the subglobose to globose basidiospores (7–9 × 6–8 µm). Sequences of ITS+nLSU+mtSSU genes were used for the phylogentic analyses using maximum likelihood, maximum parsimony, and Bayesian inference methods. The three genes’ (ITS+nLSU+mtSSU) phylogenetic analysis showed that the genera *Thelephora* and *Tomentella* grouped together within the family Thelephoraceae and three new species were nested into the genus *Thelephora*, and one new species was nested into the genus *Tomentella*.

## 1. Introduction

The genera *Amaurodon* J. Schröt., *Odontia* Pers., *Pseudotomentella* Svrcek, *Thelephora* Ehrh. ex Willd., *Tomentella* Pers. ex Pat., and *Tomentellopsis* Hjortstam belong to the family Thelephoraceae Chevall. of the order Thelephorales Corner ex Oberw. and the phylum Basidiomycota R.T. Moore [1,2]. As their common morphological characteristics are resupinate and thin basidiomata, they have been recognized as resupinate thelephoroid fungi by Kõljalg [2]. Species of this group have their own typical characteristics, such as the light blue basidiomata of *Amaurodon*, the granulose or hydnoid hymenial surface of *Odontia*, the basidiospores with bifurcate warts or spines of *Pseudotomentella*, and the absence of rhizomorphs in the genus *Tomentellopsis*. However, the genera *Tomentella* and *Thelephora* have diverse and complex morphological features, such as basidiomata, with various colors and smooth to granulose surfaces, and basidiospores, with diverse shapes and ornamentations [1,2].

Fungi represent one of the most diverse groups of organisms on Earth, with an indispensable role in the processes and functioning of forest ecosystems [3]. *Thelephora* is one of the most important taxa in basidiomycetes and has been confirmed by Willdenow with *T. terrestris* Ehrh. as the type species of the family Thelephoraceae [4,5,6,7,8,9]. *Thelephora* is a fairly well-studied ectomycorrhizal basidiomycete genus with basidiocarps of various shapes; the entire genus forms ectomycorrhizal relationships with diverse plants and contributes significantly to plant health and ecosystem stability [5,10,11,12,13,14,15,16]. As mycorrhiza-formers, *Thelephora* plays a very important role in pioneer microhabitats of coniferous forests [17,18]. Acting as white rot fungus, it can also decompose dead wood [9,10,18]. Species in the genus *Tomentella* Pat. have been recognized as ectomycorrhizal (ECM) fungi since the 1980s [2,19,20], and the *Tomentella*/*Thelephora* lineage has been found to be one of the most species-rich, frequent, and abundant groups in a variety of forest ecosystems [20,21,22].

The genus *Thelephora* is characterized by the diverse shapes of basidiomatas, which are stereoid, imbricate, rosette, infundibuliform, coralloid, or resupinate; sulcate, zonate, glabrous to strigose, somewhat radially rugulose, or wrinkled abhymenial surfaces; smooth, slightly rugose to warty hymenial surfaces; monomitic, clamped hyphal systems; verruculose or echinulate basidiospore ornamentations; and the presence or absence of cystidia [7,23,24,25,26]. Taxa of *Thelephora* are associated with a variety of plants of the families Betulaceae, Casuarinaceae, Ericaceae, Fagaceae, and Pinaceae [27]. The mycelia of these species around the plant roots can help plants obtain essential minerals and water from the soil and resist diseases and drought, making significant contributions to plant health and ecosystem stability [28]. Some species possess edible and medicinal value, of which the species *T. ganbajun* M. Zang is a delicious edible fungus with high economic value in China. Recent studies have reported that the chemically active ingredients extracted from *T. ganbajun*, such as p-biphenyl phenolic compounds, polysaccharides, steroids, and fatty acids, have multiple effects, such as antioxidant activity, antitumor activity, liver protection, and immune system enhancement for humans [28,29,30,31,32].

Classification of the kingdom of fungi has been updated continuously based on the frequent inclusion of data from DNA sequences in many phylogenetic studies [33]. Based on the early adoption of molecular systematics by mycologists, both the discovery and classification of fungi, among the more basal branches of the tree, are now being achieved due to genomic analyses and environmental DNA surveys that have been conducted [34]. *Thelephora* share similar characteristics with *Tomentella*, especially in the form, size, and type of spore ornamentations [8,28,32]. Based on phylogenetic analyses using rDNA, internal transcribed spacer (ITS) region sequences showed that *Thelephora* species mixed with *Tomentella* taxa and revealed that both genera were closely related, but the research indicated that the phylogenetic analysis of ITS loci was insufficient to resolve phylogenetic relationships among closely related taxa [16,17,35,36]. Based on ITS and nLSU analyses, Vizzini et al. [37] showed that the genera *Thelephora* and *Tomentella* species did not separate into two monophyletic groups but were intermixed and formed a well-supported monophyletic clade (*Thelephora*/*Tomentella* clade). *Thelephora* and *Tomentella* are two closely related genera within the family Thelephoraceae [38]. Morphologically, both genera share some similar characteristics with a monomitic hyphal system, having clamp connections in generative hyphae and verruculose or echinulate basidiospores, but the latter differs in having resupinate, effused, and adherent basidiomata [24,27,39,40]. Phylogenetic analyses of combined ITS and nLSU datasets within basidiomycota revealed that *Thelephora* was sister to *Tomentella* nested within Thelephoraceae, while the limits between both genera were not clear [17,38,41]. The studies showed that ITS and nLSU sequences alone cannot resolve the phylogenetic relationships in this complex group of species [15,30]. In the literature, studies on wood-inhabiting fungal molecular systematics showed that the different macroscopic characteristics within the same family or even within species with the same genus have similar microscopic characteristics [42,43,44,45,46]. Vizzini et al. [37] suggested that *Thelephora* and *Tomentella* will be considered one genus, with both genera being merged in the future.

During investigations of wood-inhabiting fungi in the Yunnan–Guizhou Plateau in China, samples representing four additional species belonging to genera *Thelephora* and *Tomentella* (Thelephoraceae) were collected. To clarify the placement and relationships between the four species, we carried out a phylogenetic and taxonomic study on *Thelephora* and *Tomentella* based on the ITS+nLSU+mtSSU sequences.

## 2. Materials and Methods

### 2.1. Sample Collection and Herbarium Specimen Preparation

Fresh fruiting bodies of the fungi were collected from Dehong, Diqing, and Zhaotong of Yunnan Province, China, and the potential ECM tree partners were gymnosperm as *Abies*, *Picea*, and *Pinus yunnanensis*, and angiosperm as *Amygdalus mira*, *Betula*, *Crataegus*, and *Quercus*. The samples were photographed in situ and fresh macroscopic details were recorded. Photographs were taken by a Jianeng 80D camera (Tokyo, Japan). All of the photos were focus stacked and merged using Helicon Focus Pro 7.7.5 software. Specimens were dried in an electric food dehydrator at 40 °C [47] and then sealed and stored in an envelope bag and deposited in the herbarium of the Southwest Forestry University (SWFC), Kunming, Yunnan Province, China. Macromorphological descriptions are based on field notes and photos captured in the field and lab. The color terminology follows Petersen [48].

### 2.2. Morphology

Micromorphological data were obtained from the dried specimens when observed under a light microscope following the previous study [43,49]. The following abbreviations are used: KOH = 5% potassium hydroxide water solution, CB = Cotton Blue, CB– = acyanophilous, IKI = Melzer’s Reagent, IKI– = both inamyloid and indextrinoid, L = mean spore length (arithmetic average for all spores), W = mean spore width (arithmetic average for all spores), Q = variation in the L/W ratios between the specimens studied, and n = a/b (number of spores (a) measured from given number (b) of specimens).

### 2.3. DNA Extraction and PCR Sequencing

The EZNA HP Fungal DNA Kit (Omega Biotechnologies Co., Ltd., Kunming, China) was used to extract DNA with some modifications from the dried specimens. The nuclear ribosomal ITS region was amplified with primers ITS5 and ITS4 [50]. The PCR procedure for ITS was as follows: initial denaturation at 95 °C for 3 min, followed by 35 cycles at 94 °C for 40 s, 58 °C for 45 s, and 72 °C for 1 min and a final extension of 72 °C for 10 min. The nuclear nLSU region was amplified with primer pair LR0R and LR7 [51,52]. The PCR procedure for nLSU was as follows: initial denaturation at 94 °C for 1 min, followed by 35 cycles at 94 °C for 30 s, 48 °C for 1 min, and 72 °C for 1.5 min, and a final extension of 72 °C for 10 min. The nuclear mtSSU region was amplified with primer pair MS1 and MS2 [50]. The PCR procedure for mtSSU was as follows: initial denaturation at 94 °C for 2 min, followed by 36 cycles at 94 °C for 45 s, 52 °C for 45 s, and 72 °C for 1 min, and a final extension of 72 °C for 10 min. The PCR products were purified and directly sequenced at Kunming Tsingke Biological Technology Limited Company, Yunnan Province, China. All of the newly generated sequences were deposited in NCBI GenBank (https://www.ncbi.nlm.nih.gov/genbank/, accessed 7 November 2024) (Table 1).

### 2.4. Phylogenetic Analyses

The sequences were aligned in MAFFT version 7 [78] using the G-INS-i strategy. The alignment was adjusted manually using AliView version 1.27 [79]. Sequences of *Phellinotus neoaridus* Drechsler-Santos & Robledo retrieved from GenBank were used as the outgroup in the ITS+nLSU+mtSSU analysis (Figure 1) [56]. Sequences of *Odontia fibrosa* (Berk. and M.A. Curtis) Kõljalg retrieved from GenBank were used as the outgroup in the ITS analysis (Figure 2) [20]. Sequences of *Odontia sparsa* Yuan Yuan, Y.C. Dai & H.S. Yuan retrieved from GenBank were used as the outgroup in the ITS+nLSU analysis (Figure 3) [55].

Maximum parsimony (MP), maximum likelihood (ML), and Bayesian inference (BI) analyses were applied to the combined three datasets following a previous study [49]. All characters were equally weighted, and gaps were treated as missing data. Trees were inferred using the heuristic search option with TBR branch swapping and 1000 random sequence additions. Max-trees were set to 5000, branches of zero length were collapsed, and all parsimonious trees were saved. Clade robustness was assessed using bootstrap (BT) analysis with 1000 pseudo replicates [80]. Descriptive tree statistics—tree length (TL), composite consistency index (CI), composite retention index (RI), composite rescaled consistency index (RC), and composite homoplasy index (HI)—were calculated for each maximum parsimonious tree generated. The combined dataset was also analyzed using maximum likelihood (ML) in RAxML-HPC2 through the CIPRES Science Gateway [81]. Branch support (BS) for the ML analysis was determined by 1000 bootstrap pseudo replicates.

MrModeltest 2.3 [82] was used to determine the best-fit evolution model for each dataset for the purposes of Bayesian inference (BI), which was performed using MrBayes 3.2.7a with a GTR+I+G model of DNA substitution and a gamma distribution rate variation across sites [83]. A total of four Markov chains were run for two runs from random starting trees for 4 million generations for ITS+nLSU+mtSSU (Figure 1) and 6 million generations for ITS (Figure 2) and 10 million generations for ITS (Figure 3), with trees and parameters sampled every 1000 generations. The first quarter of all of the generations were discarded as burn-ins. A majority rule consensus tree was computed from the remaining trees. Branches were considered significantly supported if they received a maximum likelihood bootstrap support value (BS) of ≥70%, a maximum parsimony bootstrap support value (BT) of ≥70%, or a Bayesian posterior probability (BPP) of ≥0.95.

### 2.5. Pairwise Homoplasy Test

Genealogical concordance phylogenetic species recognition analysis (GCPSR) is a tool used to check significant recombinant events. The data were analyzed using SplitsTree 4 with the pairwise homoplasy Φw PHI test to determine the recombination level within closely related species [84,85,86]. The one-locus dataset ITS with closely related species was used for the analyses. PHI results lower than 0.05 (Φw < 0.05) indicate that a significant recombination is present in the dataset. The relationships between closely related taxa were visualized by constructing split graphs from the concatenated datasets using the LogDet transformation and splits decomposition options.

## 3. Results

### 3.1. Molecular Phylogeny

The ITS+nLSU+mtSSU dataset (Figure 1) comprised sequences from 37 fungal specimens representing 26 taxa. The dataset had an aligned length of 5970 characters, of which 4753 characters were constant, 370 were variable and parsimony-uninformative, and 847 were parsimony-informative. Maximum parsimony analysis yielded four equally parsimonious trees (TL = 2567, CI = 0.6373, HI = 0.3627, RI = 0.7919, and RC = 0.5047). The best model of nucleotide evolution for the ITS+nLSU+mtSSU dataset estimated and applied in the Bayesian analysis was found to be GTR+I+G. Bayesian analysis and ML analysis resulted in a similar topology as in the MP analysis. The Bayesian analysis had an average standard deviation of split frequencies = 0.004042 (BI), and the effective sample size (ESS) across the two runs is double the average ESS (avg. ESS) = 1514. The phylogram based on the ITS+nLSU+mtSSU rDNA gene regions (Figure 1) included seven genera within Thelephoraceae (Thelephorales), including *Amaurodon*, *Lenzitopsis*, *Phellodon* P. Karst, *Polyozellus* Murrill, *Thelephora*, *Tomentella*, and *Tomentellopsis*, in which three new species were nested into the genus *Thelephora* and one new species was nested into the genus *Tomentella*. The new species *Thelephora resupinata* formed a monophyletic lineage, the taxon *T. subtropica* was grouped with *T. grandinioides* C.L. Zhao & X.F. Liu, the new taxon *T. yunnanensis* was grouped closely with *T. ganbajun*, and the new species *Tomentella tenuifarinacea* formed a monophyletic lineage.

The ITS dataset (Figure 2) comprised sequences from 56 fungal specimens representing 32 taxa. The dataset had an aligned length of 685 characters, of which 337 characters were constant, 83 were variable and parsimony-uninformative, and 265 were parsimony-informative. Maximum parsimony analysis yielded one equally parsimonious tree (TL = 1063, CI = 0.4393, HI = 0.5607, RI = 0.7524, and RC = 0.3305). The best model of nucleotide evolution for the ITS dataset estimated and applied in the Bayesian analysis was found to be GTR+I+G. Bayesian analysis and ML analysis resulted in a similar topology as in the MP analysis. The Bayesian analysis had an average standard deviation of split frequencies = 0.005792 (BI), and the effective sample size (ESS) across the two runs is double the average ESS (avg. ESS) = 711. The phylogenetic tree (Figure 2) showed that the new species *Thelephora resupinata* was the sister to *T. dominicana* Angelini, Losi & Vizzini. Furthermore, the species *T. subtropica* was grouped with the clade including three taxa, namely *T. lacunosa* Yan C. Li & Zhu L. Yang, *T. petaloides* Yan C. Li & Zhu L. Yang, and *T. sikkimensis* K. Das, Hembrom & Kuhar. Moreover, *T. yunnanensis* was grouped closely with two taxa, namely *T. palmata* (Scop.) Fr. and *T. regularis* Schwein.

The ITS+nLSU dataset (Figure 3) comprised sequences from 116 fungal specimens representing 69 taxa. The dataset had an aligned length of 2120 characters, of which 1665 characters were constant, 97 were variable and parsimony-uninformative, and 358 were parsimony-informative. Maximum parsimony analysis yielded 36 equally parsimonious trees (TL = 2461, CI = 0.2731, HI = 0.7269, RI = 0.5787, and RC = 0.1580). The best model of nucleotide evolution for the ITS+nLSU dataset estimated and applied in the Bayesian analysis was GTR+I+G. Bayesian analysis and ML analysis resulted in a topology similar to that of the MP analysis. The Bayesian analysis had an average standard deviation of split frequencies = 0.009572 (BI), and the effective sample size (ESS) across the two runs is double the average ESS (avg. ESS) = 318. The phylogenetic tree (Figure 3) showed that the new taxon *Tomentella tenuifarinacea* was found to be the sister to *T. subtestacea* (Bourdot & Galzin) Svrček.

The application of the PHI test to the ITS tree–locus sequences revealed no recombination level within phylogenetically related species. No significant recombination events were observed between *Thelephora resupinata* and *T. subtropica* and phylogenetically closely related species viz. *T. aurantiotincta*, *T. dominicana, T. grandinioides, T. lacunosa*, *T. petaloides*, and *T. sikkimensis* (Figure 4). The test results of the ITS sequence dataset show Φw = 0.0637 (Φw > 0.05) and that no recombination is present in the two new species with *T. dominicana* and *T. lacunosa.* No significant recombination events were observed between *Thelephora yunnanensis* and phylogenetically closely related species viz. *T. regularis* and *T. palmata* (Figure 5). The test results of the ITS sequence dataset show Φw = 0.1371 (Φw > 0.05) and that no recombination is present in the new species with *T. regularis* and *T. palmata*. No significant recombination events were observed between *Tomentella tenuifarinacea* and phylogenetically closely related species viz. *T. galzinii, T. pulvinulata, T. subtestacea*, and *T. viridula* (Figure 6). The test results of the ITS sequence dataset show Φw = 0.9744 (Φw > 0.05) and that no recombination is present in the new species with *T. subtestacea.*

### 3.2. Taxonomy

***Thelephora resupinata*** X.J. Zhang & C.L. Zhao, sp. nov. (Figure 7 and Figure 8).

MycoBank no.: 854533

**Holotype**—China. Yunnan Province, Diqing, Weixi County, Weideng Town, Songpo Village, GPS coordinates: 27°10′ N, 99°18′ E, altitude 2865 m.a.s.l., on the fallen branch of angiosperm, leg. C.L. Zhao, 13 November 2023, CLZhao 34,538 (SWFC).

**Etymology*—Resupinata*** (Lat.): referring to the resupinate basidiomata.

**Fruiting body**—Basidiomata annual, resupinate, coriaceous, closely adnate, up to 8 cm long, 4.5 cm wide, 1 mm thick. Hymenial surface tuberculate, cream to smoke gray when fresh, pale mouse gray to black when dry. Context fleshy tough in fresh condition, coriaceous in dried condition. Without odor when fresh and dry.

**Hyphal structure**—Hyphal system monomitic, generative hyphae with clamp connections, yellowish-brown, slightly thick-walled, moderately branched, interwoven, 5–5.5 µm in diameter. IKI–, CB–, brown-black to black in KOH.

**Hymenium**—Cystidia and cystidioles absent. Basidia barrel-shaped, with 4 sterigmata and a basal clamp, 27–36 × 11–13.5 µm. Basidioles dominant, slightly smaller than basidia.

**Spores**—Basidiospores subglobose to globose, nodulose to verrucose, yellowish-brown, thick-walled, IKI–, CB–, (8–)9–12 × (6–)7–9(–10) µm, L = 10.72 µm, W = 8.31 µm, Q = 1.26–1.29 (*n* = 60/2).

**Additional specimen (paratype) examined**—China. Yunnan Province, Diqing, Weixi County, Weideng Town, Songpo Village, GPS coordinates: 27°10′ N, 99°18′ E, altitude 2865 m.a.s.l., on the fallen branch of angiosperm, leg. C.L. Zhao, 13 November 2023, CLZhao 34548 (SWFC).

***Thelephora subtropica*** X.J. Zhang & C.L. Zhao, sp. nov. (Figure 9 and Figure 10).

MycoBank no.: 854532

**Holotype**—China. Yunnan Province, Dehong, Yingjiang County, Tongbiguan Provincial Nature Reserve, GPS coordinates: 24°49′ N, 98°61′ E, altitude 1500 m.a.s.l., on the ground of angiosperm forest, leg. C.L. Zhao, 19 July 2023, CLZhao 30590 (SWFC).

**Etymology—*Subtropica*** (Lat.): referring to distribution (subtropical zone) for the holotype of the new species.

**Fruiting body**—Basidiomata annual, solitary. Pilei medium-sized, coriaceous, infundibuliform, up to 5.5 cm long, 4.5 cm wide, 1 mm thick, buff to salmon when fresh, gray-brown when dry, proliferous from a central common base, usually with several to many laterally confluent spathulate to flabelliform, uplifted, the surface radially cream striate, margin thin, serrulate. Hymenial surface grandinoid, buff-yellow to cinnamon-buff when fresh, cinnamon-buff to peach when dry. Stipe flatted or broadened, up to 4 cm long, up to 2 cm in diameter. Context fleshy tough in fresh condition, cotton in dried condition, up to 0.3 mm thick at the thickest portion of pileus, thinner at margin and thicker towards the base. Odor strong when fresh, somewhat with the beef jerky flavor.

**Hyphal structure**—Hyphal system monomitic, generative hyphae with clamp connections, colorless, slightly thick-walled, frequently branched, loosely interwoven, 3–4.5 µm in diameter. IKI–, CB–, brown-black to black in KOH.

**Hymenium**—Cystidia and cystidioles absent. Basidia subclavate, with 4 sterigmata and a basal clamp, 28–45 × 7.5–9.5 µm. Basidioles subclavate, slightly smaller than basidia.

**Spores**—Basidiospores subglobose to globose, nodulose to verrucose, yellowish-brown, thick-walled, IKI–, CB–, 6–8(–9) × (4.5–)5–7.5 µm, L = 7.19 µm, W = 6.92 µm, Q = 1.04–1.19 (*n* = 60/2).

**Additional specimen (paratype) examined**—China. Yunnan Province, Dehong, Yingjiang County, Tongbiguan Provincial Nature Reserve, GPS coordinates: 24°49′ N, 98°61′ E, altitude 1500 m.a.s.l., on the ground of angiosperm forest, leg. C.L. Zhao, 20 July 2023, CLZhao 30591 (SWFC).

***Thelephora yunnanensis*** X.J. Zhang & C.L. Zhao, sp. nov. (Figure 11 and Figure 12).

MycoBank no.: 854531

**Holotype**—China. Yunnan Province, Zhaotong, Qiaojia County, Yaoshan National Nature Reserve, GPS coordinates: 26°90′ N, 102°95′ E, altitude 2500 m.a.s.l., on the ground of the gymnosperm forest, leg. C.L. Zhao, 24 August 2020, CLZhao 20929 (SWFC).

**Etymology—*Yunnanensis*** (Lat.): referring to the locality (Yunnan Province) of the type specimen.

**Fruiting body**—Basidiomata annual, laterally stipitate, gregarious. Pilei small to medium-sized, coriaceous, infundibuliform, up to 2.5 cm long, 2 cm wide, 1 mm thick; salmon when fresh, cinnamon-buff to gray-brown when dry, proliferous from a central common base, usually with several to many laterally confluent spathulate to flabelliform, uplifted, the surface radially black striate, margin thin, serrulate. Hymenial surface smooth, umber to coffee when fresh, coffee on drying. Stipe cylindrical, up to 2 cm long, up to 5 mm in diameter. Context fleshy tough in fresh condition, leathery in dried condition, up to 0.5 mm thick at the thickest portion of pileus, thinner at margin and thicker towards the base, gray-brown. Odor mild when fresh, somewhat with the beef jerky flavor.

**Hyphal structure**—Hyphal system monomitic, generative hyphae colorless, slightly thick-walled, with clamp connections, branched, interwoven, 5–6 µm in diameter, IKI–, CB–, brown-black to black in KOH.

**Hymenium**—Cystidia and cystidioles absent. Basidia barreled, with 4 sterigmata and a basal clamp connection, 29–35 × 9–11 µm. Basidiole slightly smaller than basidia.

**Spores**—Basidiospores subglobose to globose, yellowish-brown, nodulose to verrucose, thick-walled, IKI–, CB–, 7–10(–11.5) × (5–)6–8(–9) µm, L = 8.4 µm, W = 7 µm, Q = 1.21–1.31 (*n* = 90/3).

**Additional specimens (paratypes) examined**—China. Yunnan Province, Qiaojia Count, Yaoshan National Nature Reserve, GPS coordinates: 26°90′ N, 102°95′ E, altitude 2500 m.a.s.l., on the ground of the gymnosperm forest, leg. C.L. Zhao, 24 August 2020, CLZhao 20926, CLZhao 20935 (SWFC).

***Tomentella tenuifarinacea*** X.J. Zhang & C.L. Zhao, sp. nov. (Figure 13 and Figure 14). 

MycoBank no.: 855793

**Holotype**—China. Yunnan Province, Zhaotong, Wumengshan National Nature Reserve, GPS coordinates 27°33′ N, 103°72′ E, altitude 2500 m.a.s.l., on the fallen branch of angiosperm, leg. C.L. Zhao, 25 August 2023, CLZhao 31337 (SWFC).

**Etymology—*Tenuifarinacea*** (Lat.): referring to the thin basidiomata with farinaceous hymenophore.

**Fruiting body**—Basidiomata annual, resupinate, separable from the substrate, without odor or taste, farinaceous when fresh, becoming fragile upon drying, up to 8 cm long, 4 cm wide, 0.1–0.3 mm thick. Hymenial surface smooth, slightly olivaceous when fresh, olivaceous upon drying. Sterile margin narrow, slightly olivaceous, up to 1 mm wide.

**Hyphal structure**—Hyphal system monomitic, generative hyphae colorless, slightly thick-walled, with clamp connections, branched, interwoven, 5–6.5 µm in diameter, IKI–, CB–, brown-black to black in KOH.

**Hymenium**—Cystidia and cystidioles absent. Basidia cylindrical to narrowly clavate, with 4 sterigmata and a basal clamp connection, 33.5–41 × 7–9.5 µm, basidiole clavate, slightly smaller than basidia.

**Spores**—Basidiospores subglobose to globose, yellowish-brown, nodulose to verrucose, thick-walled, IKI–, CB–, 7–9 × (5.5–)6–8 µm, L = 8.04 µm, W = 6.93 µm, Q = 1.16 (*n* = 30/1).

## 4. Discussion

Many recently described wood-inhabiting fungal taxa have been reported worldwide [41,42,43,87], and in the present study, four new species of the family Thelephoraceae are reported based on a combination of morphological features and molecular evidence.

Molecular phylogenetic analyses of previous studies showed that the taxa of *Thelephora* and *Tomentella* were non-monophyletic groups, and they were intermixed in molecular phylogeny [6,8,35,37,88]. Additionally, the study based on the nLSU and ITS datasets showed that species of the genera *Thelephora* and *Tomentella* did not cluster into two distinct monophyletic groups but were intermixed, and so were the *Thelephora*/*Tomentella* clade [37]. In the present study, a phylogenetic analysis of the three sequences, ITS+nLSU+mtSSU, provided an improved resolution at the family level, showing that the genera *Thelephora* and *Tomentella* grouped together, which is consistent with previous results [8,14,18,37] that three new species were nested within the genus *Thelephora*, and that one new species was nested within the genus *Tomentella*.

Phylogenetically, the phylogenetic tree (Figure 1) showed that the new species *Thelephora resupinata* clustered into the genus *Thelephora*, which formed a monophyletic lineage. Based on ITS topology (Figure 2), the present study highlighted that *T. resupinata* was found to be the sister to *T. dominicana* with strong supports. However, morphologically, *T. dominicana* is different from *T. resupinata* by the minutely velutinous to tomentose hymenial surface, colorless to medium brown hyphae, and longer basidia (42–80 × 10.4–12.8 µm) [37]. Morphologically, The species *T. glaucoflora* S.R. Yang & H.S. Yuan is distinguishable from *T. resupinata* by the radial rugulose, zonate, violet-gray hymenial surface, and longer basidia measuring 40–60 × 6–10 µm [46]. The species *T. nebula* S.R. Yang & H.S. Yuan is delimited from *T. resupinata* by its grayish brown to brown hymenium surface and longer basidia measuring 45–65 × 8–11 µm [46].

Phylogenetically, the phylogenetic tree (Figure 1) showed that *Thelephora subtropica* was grouped with *T. grandinioides*. However, morphologically, *T. grandinioides* can be delimited from *T. subtropica* by having the buff to clay–buff hymenial surface, two types of cystidia (tubular cystidia and septated cystidia), and narrower basidia (27–62 × 5–7.5 µm) [60]. Based on the ITS topology (Figure 2), *T. subtropica* was grouped with three taxa as *T. lacunosa*, *T. petaloides*, and *T. sikkimensis*. However, morphologically, *T. lacunosa* is distinguishable from *T. subtropica* by having colorless to yellowish, thin-walled generative hyphal and longer subclavate basidia (72–100 × 8–12 µm) [9]. The species *T. petaloides* is distinct from *T. subtropica* by its thin-walled generative hyphae and longer basidia measuring 53–92 × 9–12 µm [9]. The species *T. sikkimensis* can be delimited from *T. subtropica* by having thin-walled generative hyphae and longer oval to subclavate basidia (72–100 × 7.5–10 µm) [9]. Morphologically, the species *T. dactyliophora* Yan C. Li & Zhu L. Yang is delimited from *T. subtropica* by its brownish-gray to gray hymenium surface and longer basidia measuring 45–72 × 6–9 µm [9]. The species *T. nebula* S.R. Yang & H.S. Yuan is distinct from *T. subtropica* by its grayish-brown to brown hymenium surface and longer utriform to subcylindrical basidia measuring 45–65 × 8–11 µm [46].

Phylogenetically, the phylogenetic tree (Figure 1) showed that the new taxon *Thelephora yunnanensis* was grouped closely with *T. ganbajun*. However, morphologically, the taxon *T. ganbajun* is different from *T. yunnanensis* by having the usually solitary basidiomata, clavate cystidia (50–85 × 6–8 µm), the narrower utriform to subcylindrica basidia (25–55 × 6–8 µm), and shorter basidiospores (5.5–7 × 5–6 µm) [46,89]. Based on the ITS topology (Figure 2), the new taxon *T. yunnanensis* was grouped closely with two taxa as *T. palmata* and *T. regularis*. However, morphologically, *T. palmata* is different from *T. yunnanensis* by the colorless or pale brown generative hyphae, flexuous–cylindrical gloeocystidia (40–56 × 8–10 µm), and longer basidia (70–100 × 9–12 µm) [25]. In addition, *T. regularis* is distinguished from *T. yunnanensis* by the inferior, smooth, grayish vinaceous or purplish fawn hymenium and shorter fuscous purple basidiospores (6–8 × 4.5–6.5 µm) [25]. Morphologically, the species *T. grandinioides* is distinguishable from *T. yunnanensis* by the clay–buff [60]. *T. pinnatifida* Yan C. Li & Zhu L. Yang is different from *T. yunnanensis* by the gregarious to caespitose, humid and leathery basidiomatas, and longer nearly clavate basidia (47–90 × 10–11.5 µm) [9].

Phylogenetically, the phylogenetic tree (Figure 1) showed that the new species *Tomentella tenuifarinacea* clustered into the genus *Tomentella*, which formed a monophyletic lineage. Based on ITS+nLSU topology (Figure 3), the present study highlighted that *T. tenuifarinacea* was found to be the sister to *T. subtestacea*. Morphologically, *T. subtestacea* is different from *T. tenuifarinacea* by the cystidia arising from hyphae of the subhymenium, up to 60 µm long, and narrower basidia (40–55 × 5–7 µm) [37]. Morphologically, *T. cinereobrunnea* X. Lu & H.S. Yuan is distinguishable from *T. tenuifarinacea* by the grayish-brown to brown hymenial surface and narrower basidia measuring 15–35 × 4–6 µm [39]. The species *Tomentella brunneoflava* H.S. Yuan & Y.C. Dai is delimited from *T. tenuifarinacea* by its brownish-yellow hymenium surface and smaller basidia measuring 10–30 × 3–5 µm [61].

One might think that the use of morphology in species recognition will soon solve all taxonomic confusions. However, with the discovery of more new species of fungi, only morphology cannot meet the needs of taxonomy; therefore, the study of fungal molecular systematics is becoming increasingly important. In general, similar species should have similar macro and micro characteristics. However, in the present study, we found that the three new species *Thelephora resupinata*, *T. subtropica,* and *T. yunnanensis* of *Thelephora* had different macroscopic characteristics within the same genus. Traditionally, the form of basidiocarps is the most important characteristic in distinguishing *Thelephora* and *Tomentella*, which are resupinate in *Tomentella*, but erect, with varied forms, to partially resupinate in *Thelephora* [18,25,37,38]. The variations in basidiocarps form may also complicate the characteristics of taxa, and the results of the morphological investigations and molecular phylogenetic analyses suggested that basidiocarps reduction happened several times independently across the evolution of thelephoroid fungi [18,37,90,91]. Taxa with reduced basidiocarps should be taken into account in the diagnoses of genera for which the initial descriptions did not cover a real spectrum of polymorphism and trends of morphological rationalization in connection with the colonization of specific habitats [18,37,92]. According to molecular data, only one genus may be recognized, and *Tomentella* will be merged into *Thelephora* [39]. So, the classification issue of both genera will be resolve based on more species and sequences in the future.

In the habitat and distribution, the thelephoroid fungi have a circumglobal distribution, ranging from polar deserts [13] to tropical forests [31], but research showed that their peak diversity was observed within the boreal zone of the planet [18,25]. The specimens involved in this study were mainly collected from subtropical forests located in Dehong, Diqing, and Zhaotong of Yunnan Province, China, where the elevation is relatively high (517–4865 m) and the aphyllophoroid fungi are very rich. The forests of *Thelephora* are primarily dominated by broad-leaved trees such as Fagaceae and Pinaceae trees. As ectomycorrhizal fungi, these species may be associated with tree species of Fagaceae and/or Pinaceae [93], but they are also capable of destroying wood debris as white rot producers [8,18]. The species of *Thelephora* were a widely distributed group found on six continents except Antarctica [6,18,25,33,93], mainly distributed across Europe in Austria, Bavaria, Belgium, Denmark, Estonia, France, Georgia, Germany, Italy, the Netherlands, Norway, Poland, Russia, Slovenia, Spain, Sweden, the UK, and Ukraine [8,25,92,93,94,95,96,97]. Additionally, the most common substrata are hardwood and conifer [6]. It is also distributed in Asia (Borneo, China, Japan, India, Malaysia, Nepal, Pakistan, Philippines, Sri Lanka, and Singapore) [4,8,25,98,99,100,101,102], North America (Bahamas, Canada, Cuba, Dominican Republic, Haiti, Jamaica, Mexico, and the USA) [4,8,18,25,29,96,97,102], South America (Uruguay) [25], Oceania (Australia and Papua New Guinea) [25], and Africa (Congo and Southern Africa) [8,25]. The diversity of Thelephoraceae in China is still not well known, especially in the subtropical and tropical regions, and many recently described taxa of thelephoroid fungi are from these areas [8]. *Thelephora resupinata*, *T. subtropica*, and *T. yunnanensis* are also from the subtropics. According to an investigation, seven species are edible and four are medicinal of the genus *Thelephora* in China, with anticancer properties, the ability to treat leukemia and boost immunity, and an anti-allergic agent [51]. Our study shows that the three new *Thelephora* species and one new *Tomentella* species grow in the angiosperm and gymnosperm forests from the subtropical zone to warm-temperate zone. In addition, the three new species in *Thelephora* may have potential medicinal and edible values.

The present study found four new taxa in broad-leaved forests (Fagaceae and/or Pinaceae) mixed with coniferous trees. China is one of the most biodiverse countries in the world [103,104,105,106,107], and more Thelephoraceae species remain to be discovered here. Therefore, further studies are needed to enrich the species diversity of Thelephoraceae.

Although most Thelephorales species are resupinate (*Amaurodon*, *Odontia, Pseudotomentella*, *Tomentella*, *Tomentellopsis*), some are stipitate hydnoid (*Hydnellum*, *Phellodon*, *Sarcodon*), stipitate poroid (B*oletopsis*), stipitate smooth (*Thelephora*), or catharelloid (*Polyozellus, Thelephora*) [39,108]. Two genera, *Polyozellus* and *Pseudotomentella*, are closely related ectomycorrhizal fungi in the order Thelephorales, and based on the RPB2, mtSSU, and nrLSU and nrSSU sequence, the study provided a strong phylogenetic signal to show that *Polyozellus* and *Pseudotomentella* were grouped into *Polyozellus* clade, and the genus *Pseudotomentella* was not closely related to *Tomentella* and *Thelephora* in the phylogenetic tree [66,109].

A close phylogenetic relationship between *Thelephora* Ehrh. ex Willd. characterized by erect, more or less branched sporocarps, and the strictly resupinate genus *Tomentella* Pers. ex Pat. has been suspected for a long time based on micromorphological features [18,25,37]. This lineage was globally one of the most species-rich ectomycorrhizal groups present in eDNA studies [55,110,111], where the separation of *Tomentella* and *Thelephora* was causing issues in the communication of taxa on the genus level since most studies used formal names instead of non-formal lineage names. Based on previous studies and yet to be published work, Kõljalg [112] proposed to merge both genera *Thelephora* and *Tomentella* species into genus *Thelephora* priority according to the nomenclatural rules, in which *Thelephora* was validly introduced a 100 years later.

In the fungal kingdom, ectomycorrhizal (EcM) symbiosis has evolved independently in multiple groups that are referred to as lineages, and a growing number of molecular studies in the fields of mycology, ecology, soil science, and microbiology generate vast amounts of sequence data from genera *Thelephora* and *Tomentella* in their natural habitats, particularly from soil and roots [113]. This study synthesized the phylogenetic and taxonomic breadth of EcM fungi by using the wealth of accumulated sequence data of *Thelephora* and *Tomentella* and compiled available information about exploration types of 143 genera of EcM fungi (including 67 new reports) that can be tentatively used to help infer the ecological strategies of different fungal groups of *Thelephora* and *Tomentella*, in which they suggested that EcM symbiosis has arisen independently in 78–82 fungal lineages that comprise 251–256 genera [113].

## Figures and Tables

**Figure 1 jof-10-00775-f001:**
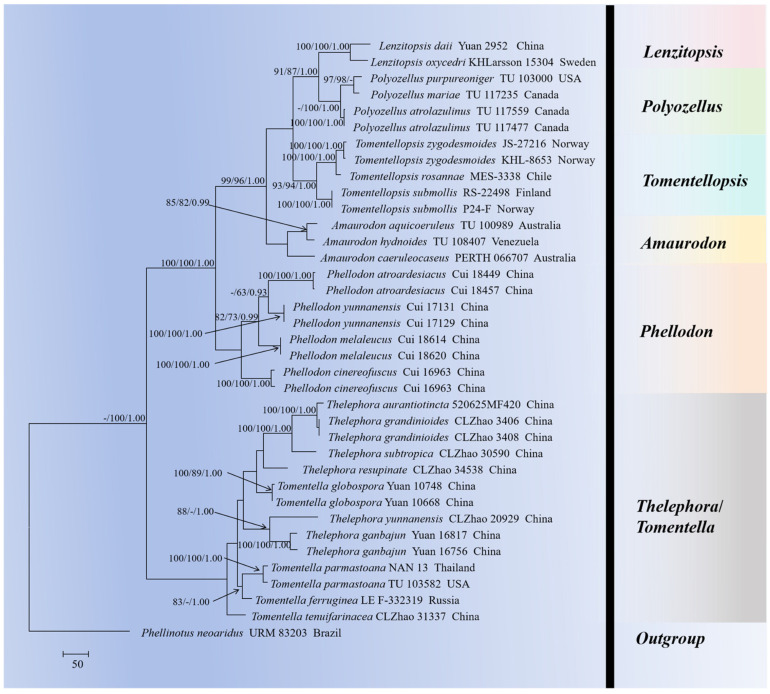
Maximum parsimony strict consensus tree illustrating the phylogeny of *Thelephora* and *Tomentella* and related genera in the family Thelephoraceae based on ITS+nLSU+mtSSU sequences. Branches are labeled with maximum likelihood bootstrap values ≥ 70%, parsimony bootstrap values ≥ 50%, and Bayesian posterior probabilities ≥ 0.95, respectively.

**Figure 2 jof-10-00775-f002:**
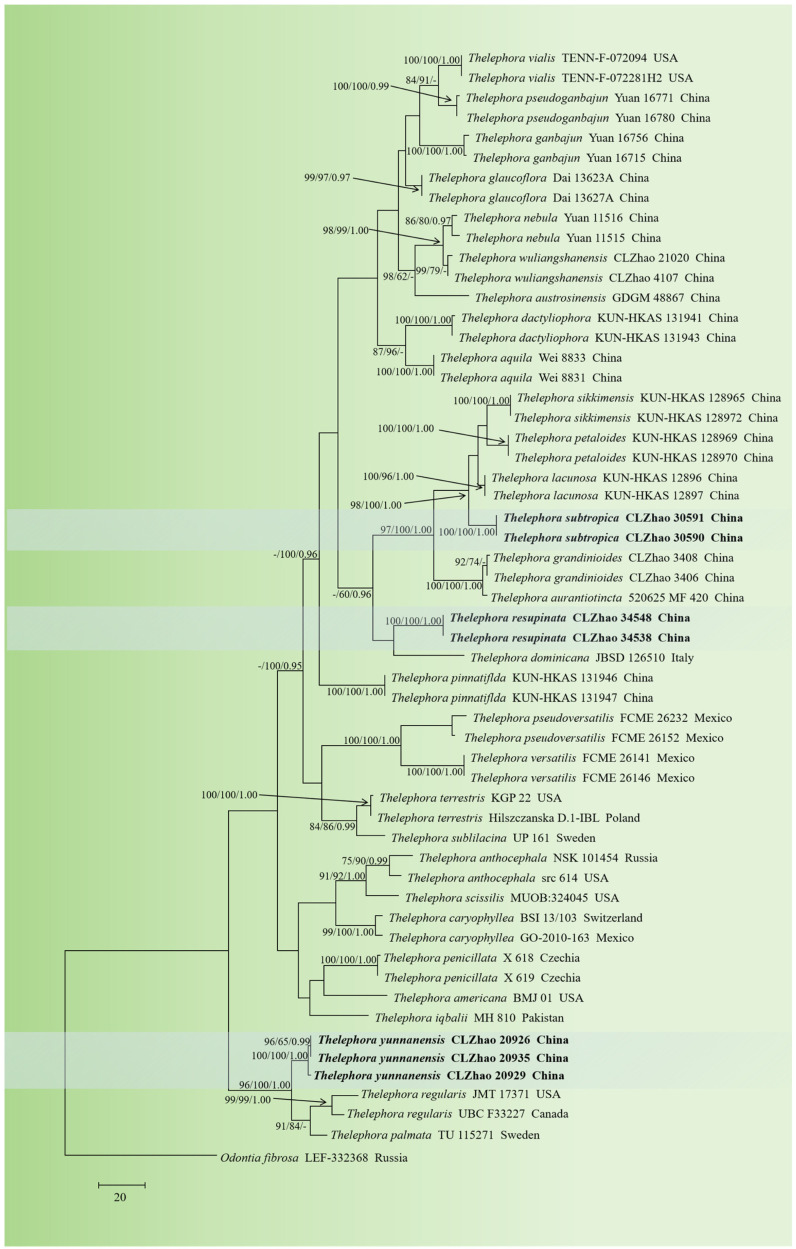
Maximum parsimony strict consensus tree illustrating the phylogeny of the three new species and related species in *Thelephora* based on ITS sequences. Branches are labeled with maximum likelihood bootstrap values ≥ 70%, parsimony bootstrap values ≥ 50%, and Bayesian posterior probabilities ≥ 0.95, respectively. The new species are in bold.

**Figure 3 jof-10-00775-f003:**
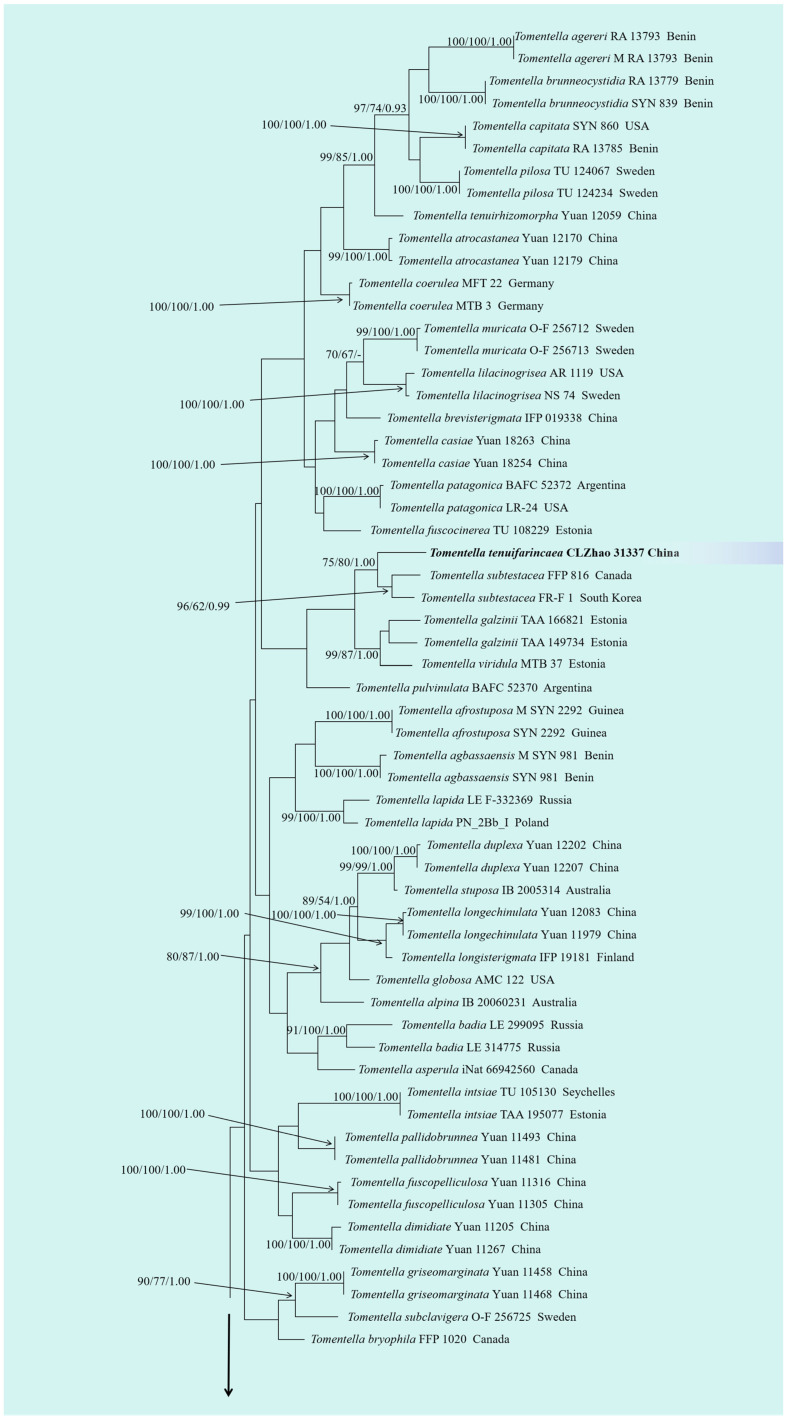
Maximum parsimony strict consensus tree illustrating the phylogeny of the one new species and related species in *Tomentella*, based on ITS+nLSU sequences. Branches are labeled with maximum likelihood bootstrap values ≥ 70%, parsimony bootstrap values ≥ 50%, and Bayesian posterior probabilities ≥ 0.95, respectively. The new species are in bold.

**Figure 4 jof-10-00775-f004:**
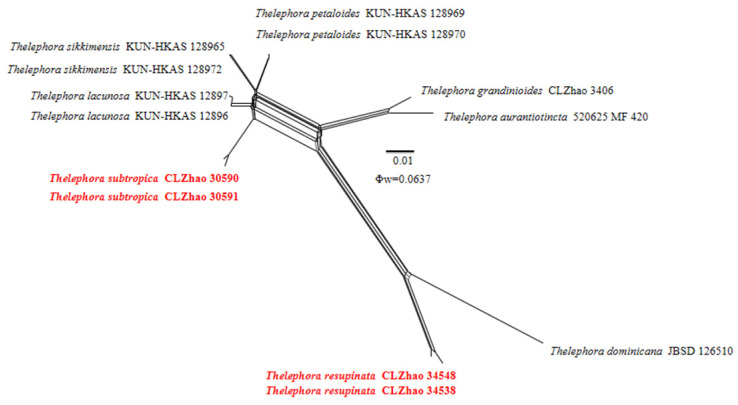
Split graphs showing the results of PHI test for the ITS data of *Thelephora resupinata* and *T. subtropica* and closely related taxa using LogDet transformation and splits decomposition. PHI test results Φw ≤ 0.05 indicate that there is significant recombination within the dataset. New taxa are in red.

**Figure 5 jof-10-00775-f005:**
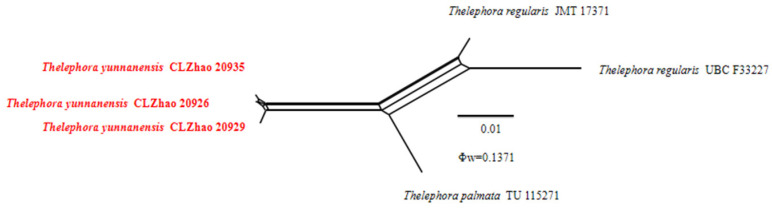
Split graphs showing the results of PHI test for the ITS data of *Thelephora yunnanensis* and closely related taxa using LogDet transformation and splits decomposition. PHI test results Φw ≤ 0.05 indicate that there is significant recombination within the dataset. New taxa are in red.

**Figure 6 jof-10-00775-f006:**
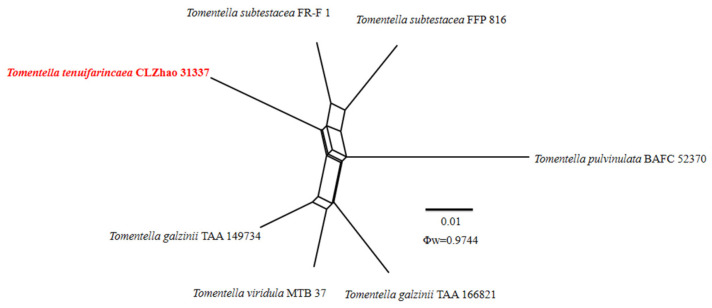
Split graphs showing the results of PHI test for the ITS data of *Tomentella tenuifarinacea* and closely related taxa using LogDet transformation and splits decomposition. PHI test results Φw ≤ 0.05 indicate that there is significant recombination within the dataset. New taxa are in red.

**Figure 7 jof-10-00775-f007:**
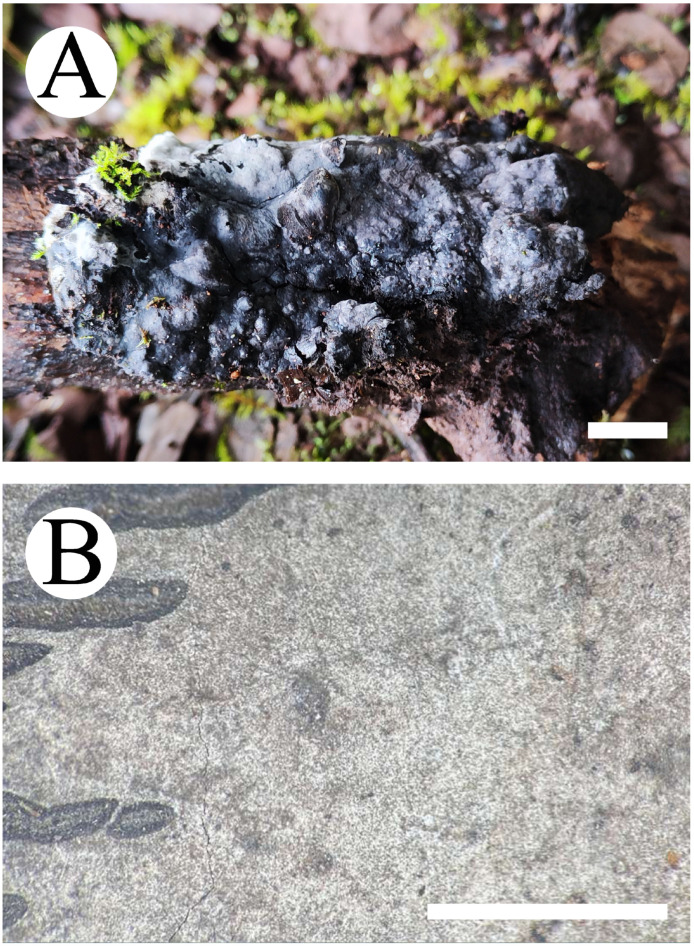
Basidiomata of *Thelephora resupinata*: CLZhao 34538 (holotype). Basidiomata on the substrate (**A**), macroscopic characteristics of hymenophore (**B**). Bars: (**A**) = 1 cm; (**B**) = 1 mm.

**Figure 8 jof-10-00775-f008:**
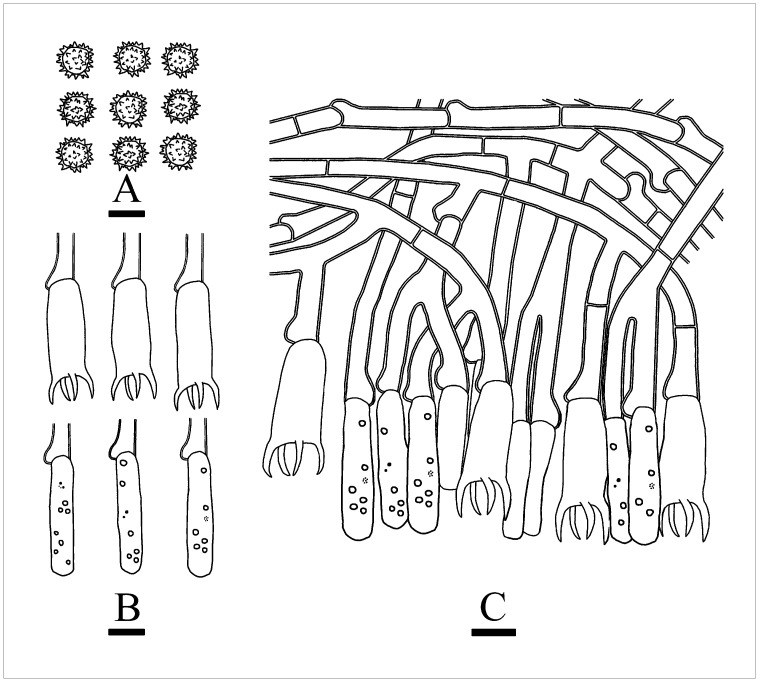
Microscopic structures of *Thelephora resupinata*: CLZhao 34538 (holotype). (**A**) Basidiospores, (**B**) basidia and basidioles, and (**C**) a section of hymenium. Bars: (**A**–**C**) = 10 µm.

**Figure 9 jof-10-00775-f009:**
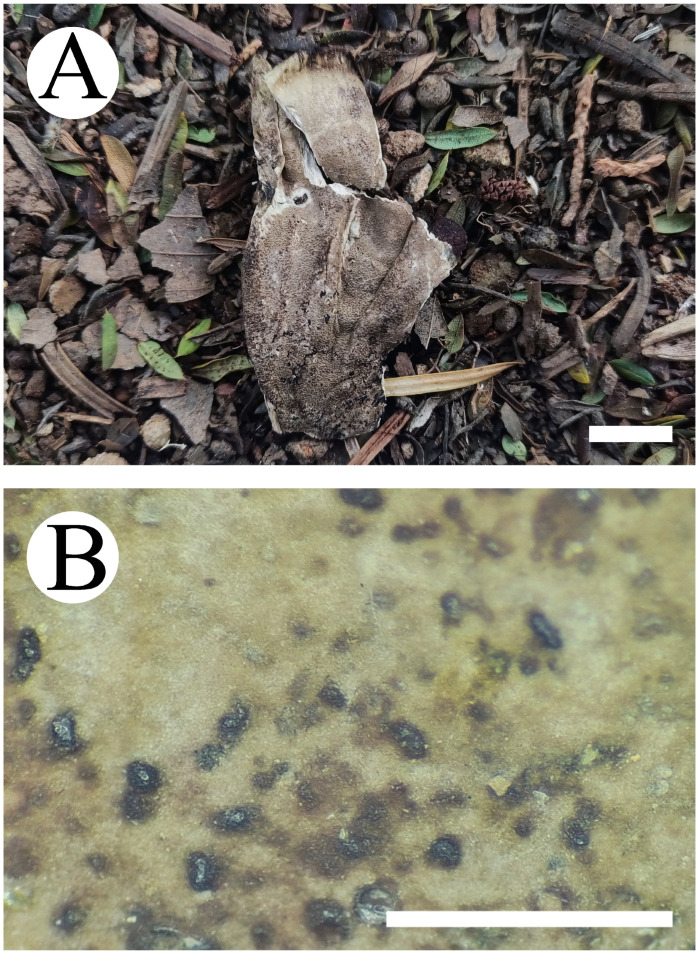
Basidiomata of *Thelephora subtropica*: CLZhao 30590 (holotype). Basidiomata on the substrate (**A**), macroscopic characteristics of hymenophore (**B**). Bars: (**A**) 1 cm; (**B**) 1 mm.

**Figure 10 jof-10-00775-f010:**
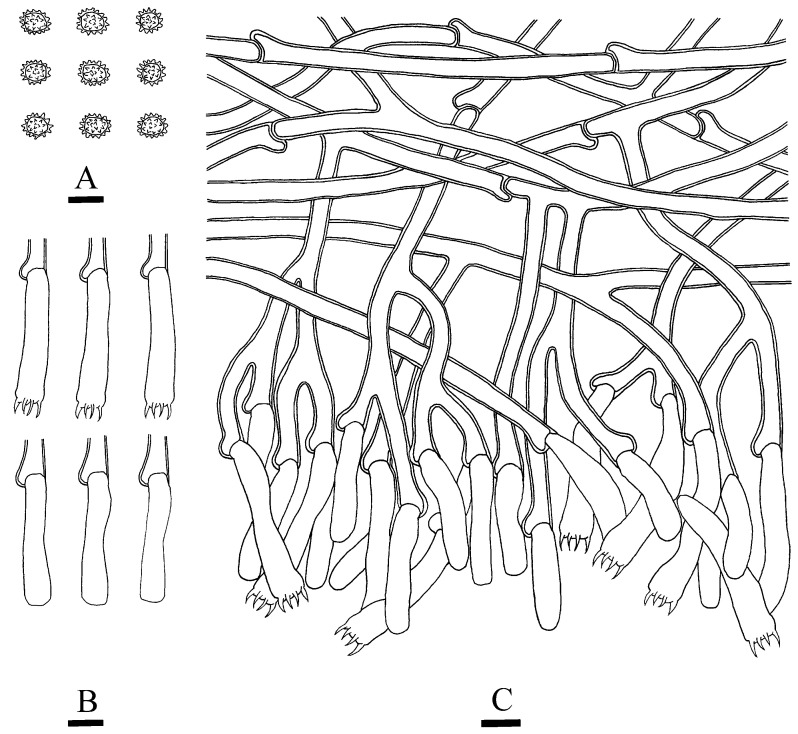
Microscopic structures of *Thelephora subtropica*: CLZhao 30590 (holotype). (**A**) Basidiospores, (**B**) basidia and basidioles, and (**C**) a section of hymenium. Bars: (**A**–**C**) = 10 µm.

**Figure 11 jof-10-00775-f011:**
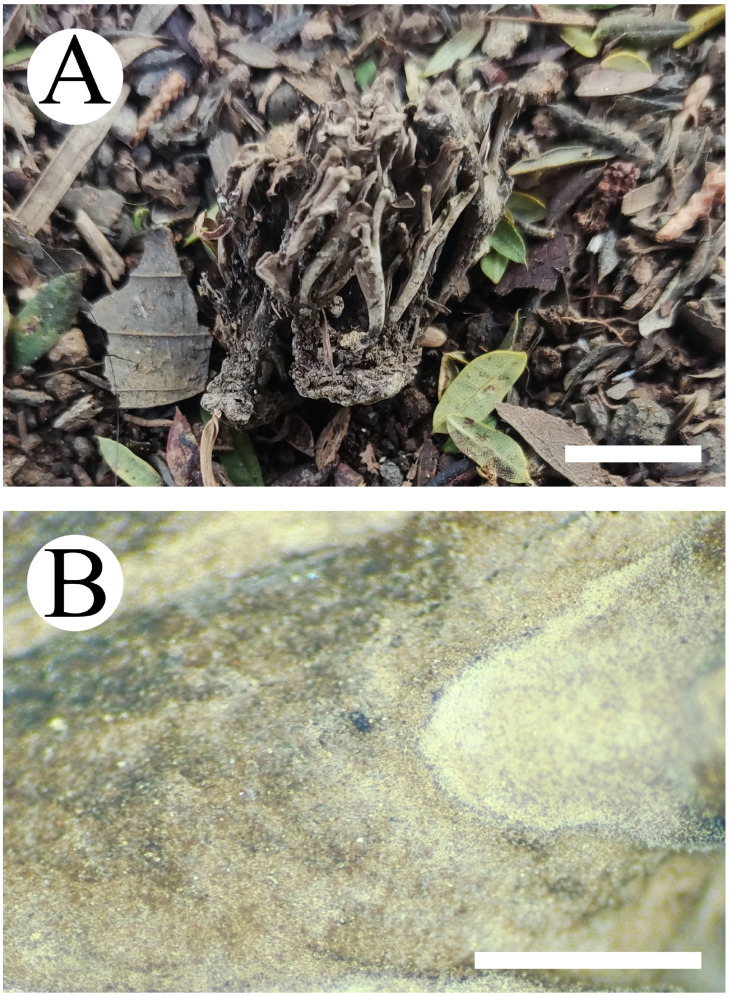
Basidiomata of *Thelephora yunnanensis*: CLZhao 20929 (holotype). Basidiomata on the substrate (**A**), macroscopic characteristics of hymenophore (**B**). Bars: (**A**) 1 cm; (**B**) 1 mm.

**Figure 12 jof-10-00775-f012:**
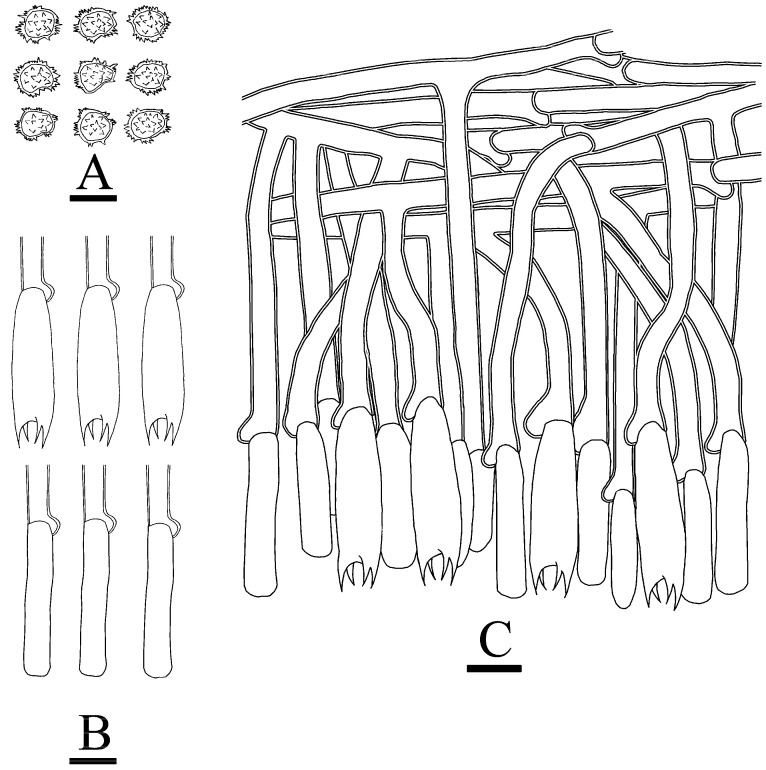
Microscopic structures of *Thelephora yunnanensis*: CLZhao 20929 (holotype). (**A**) Basidiospores, (**B**) basidia and basidioles, and (**C**) a section of hymenium. Bars: (**A**–**C**) = 10 µm.

**Figure 13 jof-10-00775-f013:**
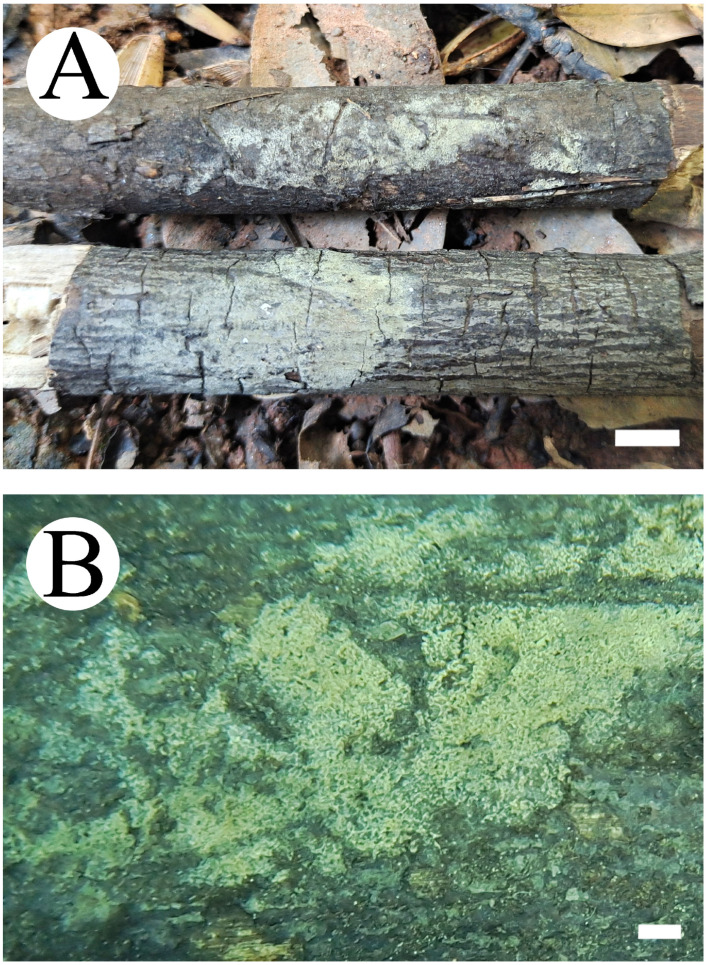
Basidiomata of *Tomentella tenuifarinacea*: CLZhao 31337 (holotype). Basidiomata on the substrate (**A**), macroscopic characteristics of hymenophore (**B**). Bars: (**A**) 1 cm; (**B**) 1 mm.

**Figure 14 jof-10-00775-f014:**
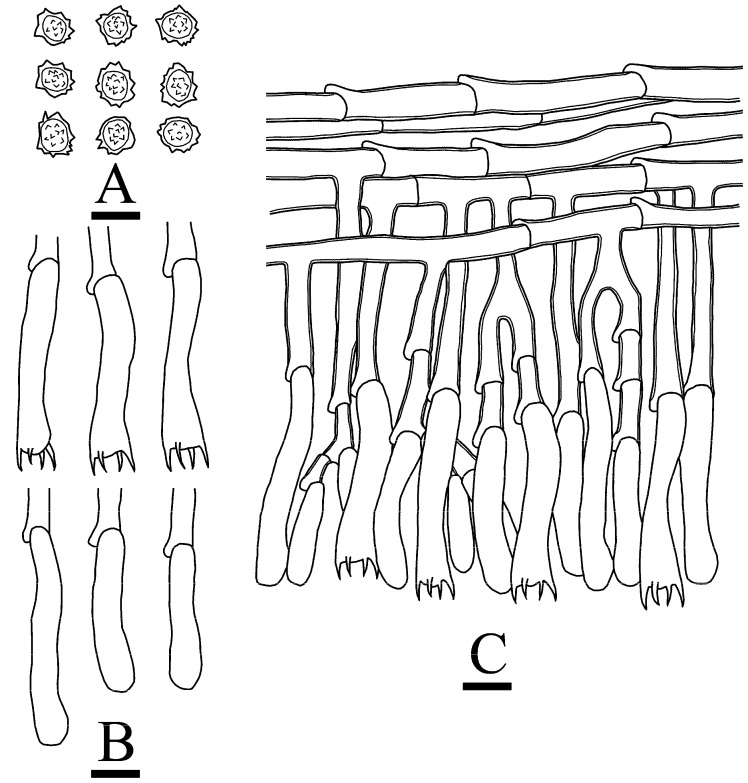
Microscopic structures of *Tomentella tenuifarinacea*: CLZhao 31337 (holotype). (**A**) Basidiospores, (**B**) basidia and basidioles, (**C**) and a section of hymenium. Bars: (**A**–**C**) = 10 µm.

**Table 1 jof-10-00775-t001:** List of species, specimens, and GenBank accession numbers of sequences used in this study. New species are shown in bold.

Species Name	Sample No.	GenBank Accession No.			Country	References
		ITS	nLSU	mtSSU		
*Amaurodon aquicoeruleus*	TU100989	AM490944	—	—	Australia	[53]
*Amaurodon caeruleocaseus*	PERTH:066707	MT565478	—	—	Australia	[53]
*Amaurodon hydnoides*	TU108407	AM490941	—	—	Venezuela	[53]
*Lenzitopsis daii*	Yuan2952	JN169798	—	—	China	[54]
*Lenzitopsis oxycedri*	KHLarsson15304	MK602774	—	—	Sweden	[54]
*Odontia fibrosa*	LE F-332368	MT981502	MT981502	—	Russia	[20]
*Odontia sparsa*	Yuan 10718	MG719980	—	—	China	[55]
*Phellinotus neoaridus*	URM 83203	MZ92858	MZ964977	—	Brazil	[56]
*Phellodon atroardesiacus*	Cui 18449	MZ221189	MZ225598	MZ225636	China	[57]
*Phellodon atroardesiacus*	Cui 18457	MZ225577	MZ225599	—	China	[57]
*Phellodon cinereofuscus*	Cui 16962	MZ225583	MZ225605	MZ225643	China	[57]
*Phellodon cinereofuscus*	Cui 16963	MZ225584	MZ225606	MZ225644	China	[57]
*Phellodon melaleucus*	Cui 18614	OL449262	OL439032	OL439022	China	[57]
*Phellodon melaleucus*	Cui 18620	OL449263	OL439033	OL439023	China	[57]
*Phellodon yunnanensis*	Cui 17129	MZ225594	MZ225614	MZ225652	China	[57]
*Phellodon yunnanensis*	Cui 17131	MZ225595	MZ225615	MZ225653	China	[57]
*Polyozellus atrolazulinus*	TU117477	MF100839	—	—	Canada	[58]
*Polyozellus atrolazulinus*	TU117559	MG214657	—	—	Canada	[58]
*Polyozellus mariae*	TU117235	MF100826	—	—	Canada	[58]
*Polyozellus purpureoniger*	TU103000	MF100821	—	—	USA	[58]
*Thelephora americana*	BMJ01	MT196971	—	—	USA	Unpublished
*Thelephora anthocephala*	NSK101420	MT773612	—	—	Russia	Unpublished
*Thelephora anthocephala*	src614	DQ974771	—	—	USA	Unpublished
*Thelephora aquila*	Wei 8833	OP793744	—	—	China	[46]
*Thelephora aquila*	Wei 8831	OP793743	—	—	China	[46]
*Thelephora aurantiotincta*	520625MF420	MZ057686	—	—	China	Unpublished
*Thelephora austrosinensis*	GDGM 48867	MF593265	—	—	China	[8]
*Thelephora caryophyllea*	BSI 13/103	KR606030	—	—	Switzerland	Unpublished
*Thelephora caryophyllea*	GO-2010-163	KC152242	—	—	Mexico	Unpublished
*Thelephora dactyliophora*	KUN-HKAS131941	OR940521	—	—	China	[9]
*Thelephora dactyliophora*	KUN-HKAS131943	OR940523	—	—	China	[9]
*Thelephora dominicana*	JBSD126510	KX216400	—	—	Italy	[37]
*Thelephora ganbajun*	Yuan 16756	OP793761	OP793790	OP793718	China	[46]
*Thelephora ganbajun*	Yuan16817	OP793762	OP793687	OP793721	China	[46]
*Thelephora glaucoflora*	Dai 13627A	OP793752	—	—	China	[46]
*Thelephora glaucoflora*	Dai 13623A	OP793751	—	—	China	[46]
*Thelephora grandinioides*	CLZhao 3406	MZ400677	MZ400671	—	China	[59]
*Thelephora grandinioides*	CLZhao 3408	MZ400674	MZ400672	—	China	[59]
*Thelephora iqbalii*	MH810	JX241471	—	—	Pakistan	[60]
*Thelephora lacunosa*	KUN-HKAS128966	OR512336	—	—	China	[9]
*Thelephora lacunosa*	KUN-HKAS128967	OR512337	—	—	China	[9]
*Thelephora* nebula	Yuan 11516	OP793746	—	—	China	[46]
*Thelephora nebula*	Yuan 11515	OP793745	—	—	China	[46]
*Thelephora palmata*	TU115271	MH310778	—	—	Sweden	Unpublished
*Thelephora penicillata*	X619	OL469899	—	—	Czechia	[60]
*Thelephora penicillata*	X618	OL469898	—	—	Czechia	[60]
*Thelephora petaloides*	KUN-HKAS128969	OR512332	—	—	China	[11]
*Thelephora petaloides*	KUN-HKAS128970	OR512333	—	—	China	[9]
*Thelephora pinnatiflda*	KUN-HKAS131946	OR940524	—	—	China	[9]
*Thelephora pinnatiflda*	KUN-HKAS131947	OR940525	—	—	China	[9]
*Thelephora pseudoganbajun*	Yuan 16780	OP793766	—	—	China	[46]
*Thelephora pseudoganbajun*	Yuan 16771	OP793765	—	—	China	[46]
*Thelephora pseudoversatilis*	FCME 26152	KJ462486	—	—	Mexico	Unpublished
*Thelephora pseudoversatilis*	FCME 26232	JX075890	—	—	Mexico	Unpublished
*Thelephora regularis*	JMT17371	U83485	—	—	USA	Unpublished
*Thelephora regularis*	UBC F33227	MG953966	—	—	Canada	Unpublished
** *Thelephora resupinata* **	**CLZhao 34548**	**PP810222**	**—**	**PQ060160**	**China**	**Present study**
** *Thelephora resupinata* **	**CLZhao 34538**	**PP810221**	**PP809695**	**PQ060159**	**China**	**Present study**
*Thelephora scissilis*	MUOB:324045	OK376730	—	—	USA	Unpublished
*Thelephora sikkimensis*	KUN-HKAS128972	OR512330	—	—	China	[9]
*Thelephora sikkimensis*	KUN-HKAS128965	OR512331	—	—	China	[9]
*Thelephora sublilacina*	UP161	EF493288	—	—	Sweden	Unpublished
** *Thelephora subtropica* **	**CLZhao 30591**	**PP810227**	**—**	**PQ060160**	**China**	**Present study**
** *Thelephora subtropica* **	**CLZhao 30590**	**PP810226**	**PP809695**	**PQ060159**	**China**	**Present study**
*Thelephora terrestris*	Hilszczanska D. 1-IBL	FJ532478	—	—	Poland	Unpublished
*Thelephora terrestris*	KGP22	DQ822828	—	—	USA	Unpublished
*Thelephora versatilis*	FCME26141	KJ462504	—	—	Mexico	Unpublished
*Thelephora versatilis*	FCME26146	NR12492	—	—	Mexico	Unpublished
*Thelephora vialis*	TENN-F- 072281H2	MN121029	—	—	USA	Unpublished
*Thelephora vialis*	TENN-F-072094	MN121022	—	—	USA	Unpublished
*Thelephora wuliangshanensis*	CLZhao 21020	MZ400672	—	—	China	[59]
*Thelephora wuliangshanensis*	CLZhao 4107	MZ400671	—	—	China	[59]
** *Thelephora yunnanensis* **	**CLZhao 20929**	**PP810224**	**PP809696**	**PQ060161**	**China**	**Present study**
** *Thelephora yunnanensis* **	**CLZhao 20935**	**PP810225**	**PP809697**	**PQ060162**	**China**	**Present study**
** *Thelephora yunnanensis* **	**CLZhao 20926**	**PP810223**	**—**	**—**	**China**	**Present study**
*Tomentella africana*	SYN 991	EF50722	—	—	Benin	[14]
*Tomentella africana*	M SYN 991	NR_119637	—	—	Benin	[14]
*Tomentella afrostuposa*	SYN 2292	JF520431	—	—	Guinea	[61]
*Tomentella afrostuposa*	M SYN 2292	NR_11992	—	—	Guinea	Unpublished
*Tomentella agbassaensis*	M SYN 981	NR_119638	—	—	Benin	Unpublished
*Tomentella agbassaensis*	SYN 981	EF507257	—	—	Benin	[62]
*Tomentella agereri*	RA 13793	EF538424	—	—	Benin	[62]
*Tomentella agereri*	M RA 13793	NR_119641	—	—	Benin	Unpublished
*Tomentella alpina*	IB 20060231	NR_121330	—	—	Australia	Unpublished
*Tomentella amyloapiculata*	SYN 893	EF507263	—	—	Benin	[63]
*Tomentella amyloapiculata*	M SYN 893	NR_119639	—	—	Benin	Unpublished
*Tomentella asperula*	iNat66942560	ON943290	—	—	Canada	Unpublished
*Tomentella atrocastanea*	Yuan 12170	MK211742	MK446337	—	China	[64]
*Tomentella atrocastanea*	Yuan 12179	MK211743	MK446338	—	China	[64]
*Tomentella aureomarginata*	Yuan 10683	MK211745	MK878395	—	China	[64]
*Tomentella badia*	LE 299095	MT981507	—	—	Russia	Unpublished
*Tomentella badia*	LE 314775	MT981499	—	—	Russia	Unpublished
*Tomentella botryoides*	O-F256708	MT146455	—	—	Sweden	[65]
*Tomentella botryoides*	O-F256707	MT14642	—	—	Sweden	[65]
*Tomentella brevisterigmata*	IFP 019338	NR_185567	—	—	China	Unpublished
*Tomentella brunneocystidia*	SYN 839	DQ848613	—	—	Benin	[32]
*Tomentella brunneocystidia*	RA 13779	DQ848610	—	—	Benin	[32]
*Tomentella brunneoflava*	Yuan 12162	MK211749	MK850194	—	China	[64]
*Tomentella brunneoflava*	Yuan 12161	MK211748	—	—	China	[64]
*Tomentella bryophila*	FFP1020	JQ711917	—	—	Canada	[65]
*Tomentella capitata*	SYN 860	DQ848612	—	—	USA	[32]
*Tomentella capitata*	RA 13785	DQ848611	—	—	Benin	[32]
*Tomentella casiae*	Yuan 18263	PP479638	PP486302	—	China	[62]
*Tomentella casiae*	Yuan 18254	PP479637	PP486299	—	China	[62]
*Tomentella castanea*	JW1	KC952674	—	—	Germany	Unpublished
*Tomentella cinerascens*	SS301	MT146467	—	—	Sweden	[66]
*Tomentella cinerascens*	SP72a	OQ418570	—	—	Sweden	[66]
*Tomentella coerulea*	MFT22	MK431005	—	—	Germany	Unpublished
*Tomentella coerulea*	MTB3	MN947340	—	—	Germany	Unpublished
*Tomentella dimidiata*	Yuan 11205	MK211704	MK446355	—	China	[64]
*Tomentella dimidiata*	Yuan 11267	MK211705	MK446356	—	China	[64]
*Tomentella duplexa*	Yuan 12202	MK211706	MK446357	—	China	[64]
*Tomentella duplexa*	Yuan 12207	MK211707	MK446358	—	China	[64]
*Tomentella efibulis*	Yuan 11241	MK211708	MK446361	—	China	[64]
*Tomentella efibulis*	Yuan 11329	MK211709	MK446362	—	China	[64]
*Tomentella ellisii*	src846	DQ974775	—	—	USA	[67]
*Tomentella fuscocinerea*	TU108229	GU214810	—	—	Estonia	Unpublished
*Tomentella fuscocrustosa*	Yuan 11420	MK211713	MK446367	—	China	[64]
*Tomentella fuscocrustosa*	Yuan 11399	MK211712	MK446366	—	China	[64]
*Tomentella fuscofarinosa*	Yuan 12142	MK211715	MK446369	—	China	[64]
*Tomentella fuscofarinosa*	Yuan 12125	MK211714	MK446368	—	China	[64]
*Tomentella fuscopelliculosa*	Yuan 11316	MK211717	—	—	China	[64]
*Tomentella fuscopelliculosa*	Yuan 11305	MK211716	MK446372	—	China	[64]
*Tomentella galzinii*	TAA166821	AF272932	—	—	Estonia	[67]
*Tomentella galzinii*	TAA149734	AF272928	—	—	Estonia	[67]
*Tomentella globosa*	AMC122	OP413006	—	—	USA	Unpublished
*Tomentella globospora*	Yuan 10748	MK446375	—	—	China	[64]
*Tomentella globospora*	Yuan 10668	MK446374	—	—	China	[64]
*Tomentella griseomarginata*	Yuan 11458	MK211720	MK446382	—	China	[64]
*Tomentella griseomarginata*	Yuan 11468	MK211721	MK446383	—	China	[64]
*Tomentella guineensis*	M SYN 2331	NR_119955	—	—	Guinea	[63]
*Tomentella guineensis*	SYN 2331	JF520432	—	—	Guinea	[63]
*Tomentella guiyangensis*	Yuan 18281	PP479645	PP486306	—	China	[68]
*Tomentella guiyangensis*	Yuan 18256	PP479643	PP486300	—	China	[68]
*Tomentella hjortstamiana*	TU103641	NR_121290	—	—	Seychelles	[69]
*Tomentella incrustata*	Yuan 12189	MK211723	MK446387	—	China	[64]
*Tomentella intsiae*	TAA195077	AM412296	—	—	Estonia	[70]
*Tomentella intsiae*	TU105130	NR_121286	—	—	Seychelles	[69]
*Tomentella lapida*	LE F-332369	MT981496	—	—	Russia	[71]
*Tomentella lapida*	PN_2Bb_I	JQ724049	—	—	Poland	[72]
*Tomentella larssoniana*	TU103690	AM412294	—	—	Estonia	[70]
*Tomentella larssoniana*	TU105082	NR_119738	—	—	Estonia	[71]
*Tomentella lilacinogrisea*	NS74	DQ068972	—	—	Sweden	[73]
*Tomentella lilacinogrisea*	AR1119	JX630832	—	—	USA	[74]
*Tomentella longechinulata*	Yuan 12083	MK211727	MK446394	—	China	[64]
*Tomentella longiaculeifera*	Yuan 10744		MK446391	—	China	[64]
*Tomentella longisterigmata*	IFP 19181	NR_161037	—	—	Finland	[75]
*Tomentella maroana*	M SYN 878	NR_119636	—	—	Benin	[40]
*Tomentella maroana*	SYN 878	EF507250	—	—	Benin	[40]
*Tomentella muricata*	O-F256712	MT146462	—	—	Sweden	[68]
*Tomentella muricata*	O-F256713	MT146461	—	—	Sweden	[68]
*Tomentella nitellina*	src675	DQ974778	—	—	USA	[66]
*Tomentella olivaceobasidiosa*	CLZhao 14051	PP810228	—	—	China	[76]
*Tomentella olivaceobasidiosa*	CLZhao 14056	PP810229	PP809698	—	China	[76]
*Tomentella olivaceomarginata*	Yuan 18268	PP479639	PP486303	—	China	[68]
*Tomentella pallidobrunnea*	Yuan 11493	MK211731	MK446402	—	China	[64]
*Tomentella pallidobrunnea*	Yuan 11481	MK211730	MK446401	—	China	[64]
*Tomentella pallidomarginata*	Yuan 11474	MK211733	MK446404	—	China	[64]
*Tomentella pallidomarginata*	Yuan 11404	MK211732	MK446403	—	China	[64]
*Tomentella parmastoana*	NAN13	MN075506	—	—	Thailand	Unpublished
*Tomentella parmastoana*	TU103582	NR_121289	—	—	USA	[70]
*Tomentella patagonica*	BAFC52372	NR_159018	—	—	Argentina	[36]
*Tomentella patagonica*	LR-24	MT366710	—	—	USA	Unpublished
*Tomentella pileocystidiata*	TU105068	NR_119739	—	—	Estonia	[69]
*Tomentella pileocystidiata*	TU10502	FM955845	—	—	Estonia	[69]
*Tomentella pilosa*	TU124067	MT146459	MT52521	—	Sweden	[65]
*Tomentella pilosa*	TU124234	MT146458	—	—	Sweden	[65]
*Tomentella pisoniae*	TU103671	NR_121358	—	—	USA	[70]
*Tomentella pisoniae*	TU103655	FN185986	—	—	Argentina	[36]
*Tomentella pulvinulata*	BAFC52370	NR_159017	—	—	Argentina	[36]
*Tomentella rotundata*	Yuan 18269	PP479641	PP486304	—	China	[68]
*Tomentella rotundata*	Yuan 18273	PP479642	PP486305	—	China	[68]
*Tomentella separata*	Yuan 10664	MK211737	MK850196	—	China	[64]
*Tomentella separata*	Yuan 1062	MK211736	MK850197	—	China	[64]
*Tomentella stuposa*	IB2005314	EF644117	—	—	Australia	[77]
*Tomentella subclavigera*	O-F256725	MT146460	—	—	Sweden	[65]
*Tomentella subtestacea*	FR-F10	MW546915	—	—	South Korea	Unpublished
*Tomentella subtestacea*	FFP816	JQ711878	—	—	Canada	[66]
*Tomentella tedersooi*	TU103663	NR_121359	—	—	Estonia	[70]
*Tomentella tedersooi*	TU103664	FN185989	—	—	Estonia	[70]
** *Tomentella tenuifarinacea* **	**CLZhao 31337**	**PQ276703**	**PQ276704**	**—**	**China**	**Present study**
*Tomentella tenuirhizomorpha*	Yuan 12059	MG799185	MN684327	—	China	[75]
*Tomentella tenuissima*	FK14070	KT032087	—	—	Argentina	[36]
*Tomentella tenuissima*	BAFC52369	NR_159016	—	—	USA	[69]
*Tomentella terrestris*	TAA159557	AF272911	—	—	Estonia	[69]
*Tomentella velutina*	CLZhao 25474	PP645440	PP809700	—	China	[76]
*Tomentella viridula*	MTB37	MN947374	—	—	Estonia	[69]
*Tomentella wumenshanensis*	CLZhao 33775	PP810230	PP809699	—	China	[76]
*Tomentella yunnanensis*	CLZhao 32532	PP810231	—	—	China	[76]
*Tomentellopsis rosannae*	MES-3338	MT366690	—	—	Chile	[77]
*Tomentellopsis submollis*	RS-22498	AJ410774	—	—	Finland	[77]
*Tomentellopsis submollis*	P24-F	AM086447	—	—	Norway	[77]
*Tomentellopsis zygodesmoides*	JS-27216	AJ410759	—	—	Norway	[77]
*Tomentellopsis zygodesmoides*	KHL-8653	AJ410761	—	—	Norway	[77]

## Data Availability

Publicly available datasets were analyzed in this study. These data can be found here: [https://www.ncbi.nlm.nih.gov/; https://www.mycobank.org/page/Simple%20names%20search, accessed on 07 November 2024].

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
