# Peer review of "The Diversity and Taxonomy of Thelephoraceae (Basidiomycota) with Descriptions of Four Species from Southwestern China"

_jof, 2024, doi:10.3390/jof10110775_

Round 1

Reviewer 1 Report

Thelephorace, especially geneta Tomentella, Thelephora, Pseudotomentella, and Tomentellopsis, belong to the most important ectomycorrhizal symbionts of deciduous forest ecosystems. However, all but Thelephora remains poorly studied because of resupinate sporocarps.

Here the authors proposed four new species Thelephora resupinata, Th. subtropica, Th. yunnanensis and Tomentella tenuifarinacea, based on a combination of the morphological features and molecular analyses.

I have only a few questions and remarks, as added below.

I have only a few questions and remarks, as added below.

1. Please describe the study sites, i.e. the place from whence the newly described specimens were taken, and their potential ECM tree partners.

2. Please add European records of Tomentella (e.g. https://doi.org/10.1016/j.funeco.2019.100908 ). According to my knowgledge, Tomentella species have wide range in the Nothern Hemispeare, and the same species occur in North America, Europe, and Asia.

3. There's newly published paper : Kõljalg et al. 2024, please involve this paper in the discussion.
I'm not convinced by the arguments presented by Kõljalg and colegues, but this paper should be mentioned.

Kõljalg, Saar, Svantesson, 2024. Merging the genus Tomentella with Thelephora (Fungi, Thelephorales)  Folia Cryptog. Estonica, Fasc. 61: 67–86 https://doi.org/10.12697/fce.2024.61.09

4. Please, add Pseudotomentella in the analyzes.

5. Please, add the information, which genera from symbiosis with trees, so are important for the functioning of forest ecosystems (according to Tedersoo, May, Smith 2010:  
DOI: 10.1007/s00572-009-0274-x  )

ECM phylogenetic lineages (bolded ones are mentined in this study):
- /phellodon-bankera - genera Phellodon, Bankera
- /tomentella-thelephora - genera Tomentella, Thelephora,
- /pseudotomentella - genera Pseudotomentella, Polyozellus
- /tomentellopsis - genus Tomentellopsis

Non-ectomycorrhizal Thelephoraceae:   Amaurodon, Lenzitopsis, Odontia

Author Response

  1. Please describe the study sites, i.e. the place from whence the newly described specimens were taken, and their potential ECM tree partners.
    Response: We have revised it as “Fresh fruiting bodies of the fungi were collected from Dehong, Diqing and Zhaotong of Yunnan Province, P.R. Chinaand the potential ECM tree partnerswere gymnosperm as Picea, Abies, and Pinus yunnanensis, and angiosperm as Betula, Amygdalus mira, and Crataegus” according to the reviewer's comment.

  1. Please add European records of Tomentella(e.g. https://doi.org/10.1016/j.funeco.2019.100908). According to my knowgledge, Tomentellaspecies have wide range in the Nothern Hemispeare, and the same species occur in North America, Europe, and Asia.
    Response: We have added 27 European records of Tomentella in the phylogenetic tree, in which it included 26 species according to the reviewer's comment.

  1. There's newly published paper: Kõljalg et al. 2024, please involve this paper in the discussion. I'm not convinced by the arguments presented by Kõljalg and colegues, but this paper should be mentioned. Kõljalg, Saar, Svantesson, 2024. Merging the genus Tomentellawith Thelephora(Fungi, Thelephorales) Folia Cryptog. Estonica, Fasc. 61: 67–86 https://doi.org/10.12697/fce.2024.61.09
    Response: We have added the discussion for this publication as “A close phylogenetic relationship between Thelephora Ehrh. ex Willd. characterized by erect, more or less branched sporocarps and the strictly resupinate genus Tomentella Pers. ex Pat. has been suspected for a long time, based on micromorphological features [18,25,37]. This lineage was globally one of the most species rich ectomycorrhizal groups present in eDNA studies [55,113-114], where the separation of Tomentella and Thelephora was causing issues in the communication of taxa on genus level, since most studies used formal names instead of non-formal lineage names. Based on previous studies and yet to be published work, Kõljalg [115] proposed to merge both genera Thelephora and Tomentella species into genus Thelephora priority according to the nomenclatural rules, in which Thelephora was validly introduced a 100 years later.

” according to the reviewer's comment.

  1. Please, add Pseudotomentellain the analyzes.
    Response: We have added the analyzes as “Although most Thelephorales species are resupinate (Amaurodon, Odontia, Pseudotomentella, Tomentella,Tomentellopsis), some are stipitate hydnoid (Hydnellum, Phellodon, Sarcodon), stipitate poroid (Boletopsis), stipitate smooth (Thelephora) or catharelloid (Polyozellus, Thelephora) [39,111]. Two genera Polyozellus and Pseudotomentella are closely related, ectomycorrhizal fungi in the order Thelephorales, and based on the RPB2, mtSSU and nrLSU and nrSSU sequence, the study provided a strong phylogenetic signal to show that Polyozellus and Pseudotomentella were grouped into Polyozellus clade, and the genus Pseudotomentella was not closely related to Tomentella and Thelephora in the phylogenetic tree [66,112]” according to the reviewer's comment.

  1. Please, add the information, which genera from symbiosis with trees, so are important for the functioning of forest ecosystems (according to Tedersoo, May, Smith 2010: DOI: 10.1007/s00572-009-0274-x )

Response: We have added the information of genera from symbiosis with trees as “In the fungal kingdom, the ectomycorrhizal (EcM) symbiosis has evolved independently in multiple groups that are referred to as lineages, and a growing number of molecular studies in the fields of mycology, ecology, soil science, and microbiology generate vast amounts of sequence data from genera Thelephora and Tomentella in their natural habitats, particularly from soil and roots [116].The study synthesized the phylogenetic and taxonomic breadth of EcM fungi by using the wealth of accumulated sequence data of Thelephora and Tomentella, and compiled available information about exploration types of 143 genera of EcM fungi (including 67 new reports) that can be tentatively used to help infer the ecological strategies of different fungal groups of Thelephora and Tomentella, in which they suggested that EcM symbiosis has arisen independently in 78-82 fungal lineages that comprise 251-256 genera [116]” according to the reviewer's comment.

Reviewer 2 Report

I congratulate the authors because they have done a very nice job with the micromorphology of the fungi they present. However, the paper lacks a solid conceptual basis (a common feature in many taxonomic papers). For example, what species concept are the authors using? I recall Aime et al. How to publish a new fungal species, or name, version 3.0. IMA Fungus 12, 11 (2021): “… for delimiting species, authors should provide a statement of the guiding species concept used to delimit newly proposed species (Lücking et al. 2020). Because the best species concepts to apply can vary, authors of new species should be familiar with the concept(s) that have been tested and applied to their group”.

I must clarify that the polyphasic approach is not a conceptual approach but a methodological one. It would be very good for this manuscript if the authors would include which conceptual approach is most appropriate in their delimitation of new species.

I recommend adding an extra line of evidence witn bona fide statistical species delimitation, using some coalescence method such as the GMYC model or bPTP method. The question is, aside we can see in the trees, there exist any molecular-evolutive evidence that support speciation?

Thanks to the editor and authors for allowing me to read this interesting manuscript. Overall, this is a manuscript that deserves to be published, however, some points still need to be carefully improved.

 1.        Important aspects in methods description need to be added for clarity and reproducibility:

 ·         Add details about MAFFT alignment parameters (alignment strategies: FFT-NS-2, G-INS-i, L-INS-i, etc.). It is useful to specify which alignment method was used along with any key parameters like ‘gap opening penalty’ or ‘max iterations’.

 ·         Add the number of generations or sampling frequency in MrBayes. The authors mention that Bayesian inference was performed using MrBayes, but details such as the number of generations for the Markov Chain Monte Carlo (MCMC) simulation, the burn-in period, and the sampling frequency should be included.

 ·         Were any convergence diagnostics used, such as checking the effective sample size, potential scale reduction factor, or examining trace files to assess if chains had converged? Remember that in a Bayesian analysis if convergence is not accomplished, the results have little value.

 ·         Authors should explicitly state whether phylogenies were reconstructed with concatenated loci or with individual genes. If the dataset was partitioned into multiple loci or gene regions, it's important to mention how partitions were treated in the analysis.

 ·         It would be helpful to ensure that the version of MrModeltest 2.3 is compatible with MrBayes 3.2.7a, especially if any version-specific issues are relevant to the analysis.

 ·         If there was any quality control step to remove low-quality sequences or ambiguous sites in aln (e.g., trimming poorly aligned regions or masking), this should be mentioned.

  ·         It's important to include the specific DNA extraction method or kit used, along with any modifications to the protocol. The citation used by the authors is not enough, they must describe their process for the sake of reproducibility. Describe the type of sample, buffer composition, incubation times, and steps used to purify or concentrate the DNA, Inter alia.

 ·   the methods lack details about the PCR reaction conditions e.g: volume of the reactions and concentration of reagents. The cycling conditions are of capital importance (e.g., initial denaturation time, number of cycles, annealing temperatures, extension times, final extension), the final concentrations of the primers the PCR reaction, etc.

 2.      I congratulate the authors because they have done a very nice job with the micromorphology of the fungi they present. However, the paper lacks a solid conceptual basis (a common feature in many taxonomic papers). For example, what species concept are the authors using? I recall Aime et al. How to publish a new fungal species, or name, version 3.0. IMA Fungus 12, 11 (2021): “… for delimiting species, authors should provide a statement of the guiding species concept used to delimit newly proposed species (Lücking et al. 2020). Because the best species concepts to apply can vary, authors of new species should be familiar with the concept(s) that have been tested and applied to their group”.

I must clarify that the polyphasic approach is not a conceptual approach but a methodological one. It would be very good for this manuscript if the authors would include which conceptual approach is most appropriate in their delimitation of new species.

Finally, I recommend adding an extra line of evidence to your species delimitation using some coalescence method such as the GMYC model.

Minor:

Strain CLZhao 20926 just have one marker, a single molecular marker in this era does not support the species hypothesis

Author Response

  1. Important aspects in methods description need to be added for clarity and reproducibility:

Response: We have added important aspects in methods description for clarity and reproducibility according to the reviewer's comment.

  1. Add details about MAFFT alignment parameters (alignment strategies: FFT-NS-2, G-INS-i, L-INS-i, etc.). It is useful to specify which alignment method was used along with any key parameters like ‘gap opening penalty’ or ‘max iterations’.

Response: We have added the details about MAFFT alignment parameters as “The sequences were aligned in MAFFT version 7 [82] using the G-INS-i strategy. The alignment was adjusted manually using AliView version 1.27 [83]. Sequences of Phellinotus neoaridus Drechsler-Santos & Robledo. Parmasto retrieved from GenBank was used as the outgroup in the ITS+nLSU+mtSSU analysis (Figure 1) [61]” according to the reviewer's comment.

  1. Add the number of generations or sampling frequency in MrBayes. The authors mention that Bayesian inference was performed using MrBayes, but details such as the number of generations for the Markov Chain Monte Carlo (MCMC) simulation, the burn-in period, and the sampling frequency should be included.

Response: We have added the number of generations or sampling frequency in MrBayes as “MrModeltest 2.3 [86] was used to determine the best-fit evolution model for each dataset for the purposes of Bayesian inference (BI) which was performed using MrBayes 3.2.7a with a GTR+I+G model of DNA substitution and a gamma distribution rate variation across sites [87]. A total of four Markov chains were run for two runs from random starting trees for 4 million generations for ITS+nLSU+mtSSU (Figure 1) and 6 million generations for ITS (Figure 2) and 10 million generations for ITS (Figure 3) with trees and parameters sampled every 1,000 generations. The first quarter of all of the generations were discarded as burn-ins. A majority rule consensus tree was computed from the remaining trees. Branches were considered as significantly supported if they received a maximum likelihood bootstrap support value (BS) of ≥ 70%, a maximum parsimony bootstrap support value (BT) of ≥ 70% or a Bayesian posterior probability (BPP) of ≥ 0.95”.

  1. Were any convergence diagnostics used, such as checking the effective sample size, potential scale reduction factor, or examining trace files to assess if chains had converged? Remember that in a Bayesian analysis if convergence is not accomplished, the results have little value.Authors should explicitly state whether phylogenies were reconstructed with concatenated loci or with individual genes. If the dataset was partitioned into multiple loci or gene regions, it's important to mention how partitions were treated in the analysis.It would be helpful to ensure that the version of MrModeltest 2.3 is compatible with MrBayes 3.2.7a, especially if any version-specific issues are relevant to the analysis.

Response: We have used the convergence diagnostics for partitioned into multiple loci or gene regions in the analysis.

  1. If there was any quality control step to remove low-quality sequences or ambiguous sites in aln (e.g., trimming poorly aligned regions or masking), this should be mentioned.

Response: We have the quality control to remove low quality sequences or ambiguous sites, and the sequences were aligned in MAFFT version 7 [82] using the G-INS-i strategy. The alignment was adjusted manually using AliView version 1.27 [83].

  1. It's important to include the specific DNA extraction method or kit used, along with any modifications to the protocol. The citation used by the authors is not enough, they must describe their process for the sake of reproducibility. Describe the type of sample, buffer composition, incubation times, and steps used to purify or concentrate the DNA, Inter alia.

Response: We have added the specific DNA extraction method or kit used as “The EZNA HP Fungal DNA Kit (Omega Biotechnologies Co., Ltd., Kunming, China) was used to extract DNA with some modifications from the dried specimens. The nuclear ribosomal ITS region was amplified with primers ITS5 and ITS4 [50]. The PCR procedure for ITS was as follows: initial denaturation at 95 °C for 3 min, followed by 35 cycles at 94 °C for 40 s, 58 °C for 45 s and 72 °C for 1 min, and a final extension of 72 °C for 10 min. The nuclear nLSU region was amplified with primer pair LR0R and LR7 [51–52]. The PCR procedure for nLSU was as follows: initial denaturation at 94 °C for 1 min, followed by 35 cycles at 94 °C for 30 s, 48 °C for 1 min and 72 °C for 1.5 min and a final extension of 72 °C for 10 min. The nuclear mt-SSU region was amplified with primer pair MS1 and MS2 [50]. The PCR procedure for mt-SSU was as follows: initial denaturation at 94 ◦C for 2 min, followed by 36 cycles at 94 ◦C for 45 s, 52 ◦C for 45 s and 72 ◦C for 1 min and a final extension of 72 ◦C for 10 min”.

  1. The methods lack details about the PCR reaction conditions e.g: volume of the reactions and concentration of reagents. The cycling conditions are of capital importance (e.g., initial denaturation time, number of cycles, annealing temperatures, extension times, final extension), the final concentrations of the primers the PCR reaction, etc.

Response: We have added the methods lack details about the PCR reaction conditions as “The PCR products were purified and directly sequenced at Kunming Tsingke Biological Technology Limited Company, Yunnan Province, China. All of the newly generated sequences were deposited in NCBI GenBank (https://www.ncbi.nlm.nih.gov/genbank/) (Table 1).” according to the reviewer's comment.

  1. I congratulate the authors because they have done a very nice job with the micromorphology of the fungi they present. However, the paper lacks a solid conceptual basis (a common feature in many taxonomic papers). For example, what species concept are the authors using? I recall Aime et al. How to publish a new fungal species, or name, version 3.0. IMA Fungus 12, 11 (2021): “… for delimiting species, authors should provide a statement of the guiding species concept used to delimit newly proposed species (Lücking et al. 2020). Because the best species concepts to apply can vary, authors of new species should be familiar with the concept(s) that have been tested and applied to their group”.I must clarify that the polyphasic approach is not a conceptual approach but a methodological one. It would be very good for this manuscript if the authors would include which conceptual approach is most appropriate in their delimitation of new species.

Response: Thanks for your nice instruction for the conceptual basis, which is worth deep consideration for us. The polyphasic approach is not a conceptual approach but a methodological one.

  1. Finally, I recommend adding an extra line of evidence to your species delimitation using some coalescence method such as the GMYC model.

Response: We have added the GMYC model for species delimitation using some coalescence method according to the reviewer's comment.

  1. Minor: Strain CLZhao 20926 just have one marker, a single molecular marker in this era does not support the species hypothesis.

Response: We have tried to get other markers several times, but it failed. Therefore, we are going to find other good ways to get sequences for specific specimens.

Round 2

Reviewer 2 Report

Once again I congratulate the authors for their work. The introduction was significantly improved.

The methodological additions with the handling of sequences are very clear and will help to replicate the work.

Of special importance is the Pairwise Homoplasy Test included in this review by the authors (I appreciate that they took this comment into account). The authors add a hard line of conceptual evidence on speciation: if two "populations" show evidence of significant genetic isolation, it is very likely that there is room for speciation. The absence of recombination between the tested taxa is an adequate way to address this hypothesis.

The authors have greatly improved the description of results and their discussion in conceptual terms, supporting the speciation hypothesis they propose with morphological, phylogenetic and molecular-evolutionary data. 

I just recommend checking some minor spelling details throughout the manuscript.

This version of the paper is biologically richer and makes a significant contribution to the taxonomy and biology of the group studied.

Author Response

  1. Once again I congratulate the authors for their work. The introduction was significantly improved. The methodological additions with the handling of sequences are very clear and will help to replicate the work. Of special importance is the Pairwise Homoplasy Test included in this review by the authors (I appreciate that they took this comment into account). The authors add a hard line of conceptual evidence on speciation: if two "populations" show evidence of significant genetic isolation, it is very likely that there is room for speciation. The absence of recombination between the tested taxa is an adequate way to address this hypothesis. The authors have greatly improved the description of results and their discussion in conceptual terms, supporting the speciation hypothesis they propose with morphological, phylogenetic and molecular-evolutionary data.

I just recommend checking some minor spelling details throughout the manuscript. This version of the paper is biologically richer and makes a significant contribution to the taxonomy and biology of the group studied.

Response: Thank you very much for your positive and constructive comments and suggestions on our manuscript.

The native English speaker of MDPI company reviewed it, and some improvements are following:

  1. 1.Delete “.”“,”.

Response: We have revised it.

  1. 2.Revised “colours” as “colors”.

Response: We have revised it according to the reviewer's comment.

  1. Revised “genus” as “genera”.

Response: We have revised it according to the reviewer's comment.

  1. Revised “has” as “have”.

Response: We have revised it according to the reviewer's comment.

  1. Delete “the type of the family Thelephoraceae is”.

Response: We have revised it.

  1. Revised “taxon” as “taxa”.

Response: We have revised it according to the reviewer's comment.

  1. Revised “validly published ” as “has been confirmed”.

Response: We have revised it according to the reviewer's comment.

  1. Delete “the”.

Response: We have revised it.

  1. Revised “significant contribution to” as “contributes significantly”.

Response: We have revised it.

  1. Add“of the family Thelephoraceae”.

Response: We have revised it according to the reviewer's comment.

  1. Add“a”.

Response: We have revised it according to the reviewer's comment.

  1. Revised “its” as “the”.

Response: We have revised it according to the reviewer's comment.

  1. Revised “fungi, they also” as “fungus, it can also”.

Response: We have revised it according to the reviewer's comment.

  1. Add“the”.

Response: We have revised it according to the reviewer's comment.

  1. Delete “the”.

Response: We have revised it.

  1. Revised “its” as “the”.

Response: We have revised it according to the reviewer's comment.

  1. Revised “, the”as “ ; ”.

Response: We have revised it according to the reviewer's comment.

  1. Revised “documented” as “the”.

Response: We have revised it according to the reviewer's comment.

  1. Revised “such as p-biphenyl phenolic compounds, polysaccharides, ste roids and fatty acids, extracted from T. ganbajunhave multiple effects such as antioxidant, antitumor” as “that the chemically active ingredients extracted from T. ganbajun, such as p-biphenyl phenolic compounds, polysaccharides, steroids and fatty acids, have multiple effects such as antioxidant activity, antitumor activity”.

Response: We have revised it according to the reviewer's comment.

  1. Revised “embrace” as “adoption”.

Response: We have revised it according to the reviewer's comment.

  1. Delete “the”.

Response: We have revised it.

  1. Revised “previous researches, the studies about” as “the literature, studies on”.

Response: We have revised it as “the literature, studies on” according to the reviewer's comment.

  1. Revised “significant contribution to” as “contributes significantly”.

Response: We have revised it.

  1. Revised “the” as “species with”.

Response: We have revised it according to the reviewer's comment.

  1. Revised “in” as“into”.

Response: We have revised it according to the reviewer's comment.

  1. Add“being merged”.

Response: We have revised it.

  1. Revised “of” as “in”.

Response: We have revised according to the reviewer's comment.

  1. Revised “the genus” as “species of the genera”.

Response: We have revised it according to the reviewer's comment.

  1. Revised “they are” as “were”.

Response: We have revised it according to the reviewer's comment.

  1. Add“so were”.

Response: We have revised it according to the reviewer's comment.

  1. Revised “the three genes (ITS+nLSU+mtSSU) phylogenetic analysis provided an improved resolution at the family level,showed” as “a phylogenetic analysis of the three sequences, ITS+nLSU+mtSSU, provided an improved resolution at the family level, showing”.

Response: We have revised it as “a phylogenetic analysis of the three sequences, ITS+nLSU+mtSSU, provided an improved resolution at the family level, showing” according to the reviewer's comment.

  1. Revised “into” as “within”.

Response: We have revised it according to the reviewer's comment.

  1. Revised “the” as “that”.

Response: We have revised it as “that” according to the reviewer's comment.
